# Approximating Functions with Approximate Privacy for Applications in Signal Estimation and Learning

**DOI:** 10.3390/e25050825

**Published:** 2023-05-22

**Authors:** Naima Tasnim, Jafar Mohammadi, Anand D. Sarwate, Hafiz Imtiaz

**Affiliations:** 1Department of Electrical and Electronic Engineering, Bangladesh University of Engineering and Technology, Dhaka P.O. Box 1205, Bangladesh; 0421062525@eee.buet.ac.bd; 2Nokia, Werinherstraße 91, 81541 Munich, Germany; jafar.mohammadi@nokia.com; 3Department of Electrical and Computer Engineering, Rutgers, The State University of New Jersey, 94 Brett Road, Piscataway, NJ 08854-8058, USA; ads221@soe.rutgers.edu

**Keywords:** differential privacy, functional mechanism, decentralized-data systems

## Abstract

Large corporations, government entities and institutions such as hospitals and census bureaus routinely collect our personal and sensitive information for providing services. A key technological challenge is designing algorithms for these services that provide useful results, while simultaneously maintaining the privacy of the individuals whose data are being shared. Differential privacy (DP) is a cryptographically motivated and mathematically rigorous approach for addressing this challenge. Under DP, a randomized algorithm provides privacy guarantees by approximating the desired functionality, leading to a privacy–utility trade-off. Strong (pure DP) privacy guarantees are often costly in terms of utility. Motivated by the need for a more efficient mechanism with better privacy–utility trade-off, we propose Gaussian FM, an improvement to the functional mechanism (FM) that offers higher utility at the expense of a weakened (approximate) DP guarantee. We analytically show that the proposed Gaussian FM algorithm can offer orders of magnitude smaller noise compared to the existing FM algorithms. We further extend our Gaussian FM algorithm to decentralized-data settings by incorporating the CAPE protocol and propose capeFM. Our method can offer the same level of utility as its centralized counterparts for a range of parameter choices. We empirically show that our proposed algorithms outperform existing state-of-the-art approaches on synthetic and real datasets.

## 1. Introduction

Differential privacy (DP) [1] has emerged as a de facto standard for privacy-preserving technologies in research and practice due to the quantifiable privacy guarantee it provides. DP involves randomizing the outputs of an algorithm in such a way that the presence or absence of a single individual’s information within a database does not significantly affect the outcome of the algorithm. DP typically introduces randomness in the form of additive noise, ensuring that an adversary cannot infer any information about a particular record with high confidence. The key challenge is to keep the performance or *utility* of the noisy algorithm close enough to the unperturbed one to be useful in practice [2].

In its pure form, DP measures privacy risk by a parameter ϵ, which can be interpreted as the *privacy budget*, that bounds the log-likelihood ratio of the output of a private algorithm under two datasets differing in a single individual’s data. The smaller ϵ used, the greater the privacy ensured, but at the cost of worse performance. In privacy-preserving machine learning models, higher values of ϵ are generally chosen to achieve acceptable utility. However, setting ϵ to arbitrarily large values severely undermines privacy, although there are no hard threshold values for ϵ above which formal guarantees provided by DP become meaningless in practice [3]. In order to improve utility for a given privacy budget, a relaxed definition of differential privacy, referred to as (ϵ,δ)-DP, was proposed [4]. Under this privacy notion, a randomized algorithm is considered privacy-preserving if the privacy loss of the output is smaller than exp(ϵ) with a high probability (i.e., with probability at least 1−δ) [5].

Our current work is motivated by the necessity of a decentralized differentially private algorithm to efficiently solve practical signal estimation and learning problems that (i) offers better privacy–utility trade-off compared to existing approaches, and (ii) offers similar utility as the pooled-data (or centralized) scenario. Some noteworthy real-world examples of systems that may need such differentially private decentralized solutions include [6]: (i) medical research consortium of healthcare centers and labs, (ii) decentralized speech processing systems for learning model parameters for speaker recognition, (iii) multi-party cyber-physical systems. To this end, we first focus on improving the privacy–utility trade-off of a well known DP mechanism, called the *functional mechanism (FM)* [7]. The FM approach is more general and requires fewer assumptions on the objective function than other objective perturbation approaches [8,9].

The functional mechanism was originally proposed for “pure” ϵ-DP. However, it involves an additive noise with very large variance for datasets with even moderate ambient dimension, leading to a severe degradation in utility. We propose a natural “approximate” (ϵ,δ)-DP variant using Gaussian noise and show that the proposed *Gaussian FM* scheme significantly reduces the additive noise variance. A recent work by Ding et al. [10] proposed *relaxed FM* using the Extended Gaussian mechanism [11], which also guarantees approximate (ϵ,δ)-DP instead of pure DP. However, we will show analytically and empirically that, just like the original FM, the relaxed FM also suffers from prohibitively large noise variance even for moderate ambient dimensions. Our tighter sensitivity analysis for the Gaussian FM, which is different from the technique used in [10], allows us to achieve much better utility for the same privacy guarantee. We further extend the proposed Gaussian FM framework to the decentralized or “federated” learning setting using the CAPE protocol [6]. Our capeFM algorithm can offer the same level of utility as the centralized case over a range of parameters. Our empirical evaluation of the proposed algorithms on synthetic and real datasets demonstrates the superiority of the proposed schemes over the existing methods. We now review the relevant existing research works in this area before summarizing our contributions.

**Related Works.** There is a vast literature on the perturbation techniques to ensure DP in machine learning algorithms. The simplest method for ensuring that an algorithm satisfies DP is *input perturbation*, where noise is introduced to the input of the algorithm [2]. Another common approach is *output perturbation*, which obtains DP by adding noise to the output of the problem. In many machine learning algorithms, the underlying objective function is minimized with gradient descent. As the gradient is dependent on the privacy-sensitive data, randomization is introduced at each step of the gradient descent [9,12]. The amount of noise we need to add at each step depends on the *sensitivity* of the function to changes in its input [4]. *Objective perturbation* [8,9,13] is another state-of-the-art method to obtain DP, where noise is added to the underlying objective function of the machine learning algorithm, rather than its solutions. A newly proposed take on output perturbation [14] injects noise after model convergence, which imposes some additional constraints. In addition to optimization problems, Smith [15] proposed a general approach for computing summary statistics using the *sample-and-aggregate* framework and both the Laplace and Exponential mechanisms [16].

Zhang et al. originally proposed *functional mechanism (FM)* [7] as an extension to the Laplace mechanism. FM has been used in numerous studies to ensure DP in practical settings. Jorgensen et al. applied FM in personalized differential privacy (PDP) [17], where the privacy requirements are specified at the user-level, rather than by a single, global privacy parameter. FM has also been combined with homomorphic encryption [18] to obtain both data secrecy and output privacy, as well as with fairness-aware learning [10,19] in classification models. The work of Fredrikson et al. [20], which demonstrated privacy in pharmacogenetics using FM and other DP mechanisms, is of particular interest to us. Pharmacogenetic models [21,22,23,24] contain sensitive clinical and genomic data that need to be protected. However, poor utility of differentially private pharmacogenetic models can expose patients to increased risk of disease. Fredrikson et al. [20] tested the efficacy of such models against attribute inference by using a model inversion technique. Their study shows that, although not explicitly designed to protect attribute privacy, DP can prevent attackers from accurately predicting genetic markers if ϵ is sufficiently small (≤1). However, the small value of ϵ results in poor utility of the models due to excessive noise addition, leading them to conclude that when utility cannot be compromised much, *the existing methods do not give an ϵ for which state-of-the-art DP mechanisms can be reasonably employed*. As mentioned before, Ding et al. [10] recently proposed relaxed FM in an attempt to improve upon the original FM using the Extended Gaussian mechanism [11], which offered approximate DP guarantee.

DP algorithms provide different guarantees than Secure Multi-party Computation (SMC)-based methods. Several studies [25,26,27] applied a combination of SMC and DP for distributed learning. Gade and Vaidya [25] demonstrated one such method in which each site adds and subtracts arbitrary functions to confuse the adversary. Heikkilä et al. [26] also studied the relationship of additive noise and sample size in a distributed setting. In their model, *S* data holders communicate their data to *M* computation nodes to compute a function. Tajeddine et al. [27] used DP-SMC on vertically partitioned data, i.e., where data of the same participants are distributed across multiple parties or data holders. Bonawitz et al. [28] proposed a communication-efficient method for federated learning over a large number of mobile devices. More recently, Heikkilä et al. [29] considered DP in a cross-silo federated learning setting by combining it with additive homomorphic secure summation protocols. Xu et al. [30] investigated DP for multiparty learning in vertically partitioned data setting. Their proposed framework dissects the objective function into single-party and cross-party sub-functions, and applies functional mechanisms and secure aggregation to achieve the same utility as the centralized DP model. Inspired by the seminal work of Dwork et al. [31] that proposed distributed noise generation for preserving privacy, Imtiaz et al. [6] proposed the *Correlation Private Estimation (*CAPE*)* protocol. CAPE employs a similar principle as Anandan and Clifton [32] to *reduce* the noise added for DP in decentralized-data settings.

**Our Contributions.** As mentioned before, we are motivated by the necessity of a decentralized differentially private algorithm that injects a smaller amount of noise (compared to existing approaches) to efficiently solve practical signal estimation and learning problems. To that end, we first propose an improvement to the existing functional mechanism. We achieve this by performing a tighter characterization of the sensitivity analysis, which significantly reduces the additive noise variance. As we utilize the Gaussian mechanism [33] to ensure (ϵ,δ)-DP, we call our improved functional mechanism *Gaussian FM*. Using our novel sensitivity analysis, we show that the proposed Gaussian FM injects a much smaller amount of additive noise compared to the original FM [7] and the relaxed FM [10] algorithms. We empirically show the superiority of Gaussian FM in terms of privacy guarantee and utility by comparing it with the corresponding non-private algorithm, the original FM [7], the relaxed FM [10], the objective perturbation [8], and the noisy gradient descent [12] methods. Note that the original FM [7] and the objective perturbation [8] methods guarantee pure DP, whereas the other methods guarantee approximate DP. We compare our (ϵ,δ)-DP Gaussian FM with the pure DP algorithms as a means for investigating how much performance/utility gain one can achieve by trading off pure the DP guarantee with an approximate DP guarantee. Additionally, the noisy gradient descent method is a multi-round algorithm. Due to the composition theorem of differential privacy [33], the privacy budgets in multi-round algorithms accumulate across the number of iterations during training. In order to perform better accounting for the total privacy loss in the noisy gradient descent algorithm, we use Rényi differential privacy [34].

Considering the fact that machine learning algorithms are often used in decentralized/federated data settings, we adapt our proposed Gaussian FM algorithm to decentralized/federated data settings following the (CAPE) [6] protocol, and propose capeFM. In many signal processing and machine learning applications, where privacy regulations prevent sites from sharing the local raw data, joint learning across datasets can yield discoveries that are impossible to obtain from a single site. Motivated by scientific collaborations that are common in human health research, CAPE improves upon the conventional decentralized DP schemes and achieves the same level of utility as the pooled-data scenario in certain regimes. It has been shown [6] that CAPE can benefit computations with sensitivies satisfying some conditions. Many functions of interest in machine learning and deep neural networks have sensitivites that satisfy these conditions. Our proposed capeFM algorithm utilizes the Stone–Weierstrass theorem [35] to approximate a cost function in the decentralized-data setting and employs the CAPE protocol.

To summarize, the goal of our work is to improve the privacy–utility trade-off and reduce the amount of noise in the functional mechanism at the expense of approximate DP guarantee for applications of machine learning in decentralized/federated data settings, similar to those found in research consortia. Our main contributions are:We propose Gaussian FM as an improvement over the existing functional mechanism by performing a tighter sensitivity analysis. Our novel analysis has two major features: (i) the sensitivity parameters of the data-dependent (hence, privacy-sensitive) polynomial coefficients of the Stone–Weierstrass decomposition of the objective function are free of the dataset dimensionality; and (ii) the additive noise for privacy is tailored for the *order* of the polynomial coefficient of the Stone–Weierstrass decomposition of the objective function, rather than being the same for all coefficients. These features give our proposed Gaussian FM a significant advantage by offering much less noisy function computation compared to both the original FM [7] and the relaxed FM [10], as shown for linear and logistic regression problems. We also empirically validate this on real and synthetic data.We extend our Gaussian FM to decentralized/federated data settings to propose capeFM, a novel extension of the functional mechanism for decentralized-data. To this end, we note another significant advantage of our proposed Gaussian FM over the original FM: the Gaussian FM can be readily extended to decentralized/federated data settings by exploiting the fact that the sum of a number of Gaussian random variables is another Gaussian random variable, which is not true for Laplace random variables. We show that the proposed capeFM can achieve the same utility as the pooled-data scenario for some parameter choices. To the best of our knowledge, our work is the first functional mechanism for decentralized-data settings.We demonstrate the effectiveness of our algorithms with varying privacy and dataset parameters. Our privacy analysis and empirical results on real and synthetic datasets show that the proposed algorithms can achieve much better utility than the existing state-of-the-art algorithms.

## 2. Definitions and Preliminaries

**Notation.** We denote vectors, matrices, and scalars with bold lower case letters (x), bold upper case letters (X), and unbolded letters (N), respectively. We denote indices with lower case letters and they typically run from 1 to their upper case versions (d∈1,2,…,D≜[D]). The *n*-th column of a matrix **X** is denoted as $x_{n}$. We denote the Euclidean (or $L_{2}$) norm of a vector and the spectral norm of a matrix with ${\left\|\cdot \right\|}_{2}$. Finally, we denote the inner product of two matrices **A** and **B** as $\langle A,B\rangle =\mathrm{tr}\left(A^{⊤}B\right)$.

### 2.1. Definitions

**Definition** **1**((ϵ,δ)-Differential Privacy [4])**.**
*Let us consider a domain D of datasets consisting of N records, and $D,D^{\prime }\in D$ where D and $D^{\prime }$ differ in a single record (neighboring datasets). Then, for all measurable S⊆T and all neighboring data sets $D,D^{\prime }\in D$, an algorithm A:D↦T provides (ϵ,δ)-differential privacy ((ϵ,δ)-DP) if*
$$\mathrm{Pr}\left[A\left(D\right)\in S\right]\leq \exp\left(\epsilon \right)\mathrm{Pr}\left[A\left(D^{\prime }\right)\in S\right]+\delta .$$

This definition is also known as bounded differential privacy (as opposed to unbounded differential privacy [1]). One way to interpret this is that an algorithm A satisfies (ϵ,δ)-DP if the probability distribution of the output of A does not change significantly if the input database is changed by one sample. That is to say, whether or not a particular individual takes part in a differentially private study, the outcome of the study is not changed by much. An adversary attempting to identify an individual will not be able to verify the individual’s presence or absence in the study with high confidence. The privacy of the individual is thus preserved by plausible deniability. In the definition of DP, (ϵ,δ) are privacy parameters, where lower (ϵ,δ) ensure more privacy. The parameter δ can be interpreted as the probability that the algorithm fails to provide privacy risk ϵ. Note that (ϵ,δ)-DP is known as *approximate* differential privacy whereas ϵ-differential privacy (ϵ-DP) is known as *pure* differential privacy. In general, we denote approximate (bounded) differentially private algorithms with DP. An important feature of DP is that post-processing of the output does not change the privacy guarantee, as long as that post-processing does not use the original data [33]. Among the most commonly used mechanisms for formulating a DP algorithm are additive noise mechanisms such as the Gaussian [4] or Laplace [33] mechanisms, and random sampling using the exponential mechanism [16]. For additive noise mechanisms, the standard deviation of the additive noise is scaled to the *sensitivity* of the computation.

**Definition** **2**($L_{p}$-Sensitivity [4])**.**
*Given neighboring datasets D and $D^{\prime }$, the $L_{p}$-sensitivity of a vector-valued function f(D) is*
$$\Delta \max_{D,D^{\prime }}{\left\|f\left(D\right)-f\left(D^{\prime }\right)\right\|}_{p}.$$

We focus on p=1 and 2 in this paper.

**Definition** **3**(Gaussian Mechanism [33]). *Let $f:D\mapsto R^{D}$ be an arbitrary function with $L_{2}$-sensitivity *Δ*. The Gaussian mechanism with parameter τ adds noise scaled to $N(0,\tau^{2})$ to each of the D entries of the output and satisfies (ϵ,δ)-differential privacy for ϵ∈(0,1) if*
$$\tau \geq \frac{\Delta }{\epsilon }\sqrt{2\log\frac{1.25}{\delta }}.$$

Note that, for any given (ϵ,δ) pair, we can calculate a noise variance $\tau^{2}$ such that addition of a noise term drawn from $N(0,\tau^{2})$ guarantees (ϵ,δ)-differential privacy. There are infinitely many (ϵ,δ) pairs that yield the same $\tau^{2}$. Therefore, we parameterize our methods using $\tau^{2}$ [36] in this paper. We refer the reader to [37,38,39] for a broader discussion of privacy parameter ϵ.

**Definition** **4**(Rényi Differential Privacy (RDP) [34])**.**
*A randomized mechanism A:D↦T is $\left(a,\epsilon_{r}\right)$-Rényi differentially private if, for any adjacent $D,D^{\prime }\in D$, the following holds:*
$$D_{a}\left(A\left(D\right)\|A\left(D^{\prime }\right)\right)\leq \epsilon_{r}$$
*Here, $D_{a}\left(P\left(x\right)\|Q\left(x\right)\right)=\frac{1}{a-1}\log E_{x∼Q}{\left(\frac{P(x)}{Q(x)}\right)}^{a}$, and P(x) and Q(x) are probability density functions defined on T.*

Analyzing the total privacy loss of a multi-round algorithm, each stage of which is DP, is a challenging task. It has been shown [34,40] that the advanced composition theorem [33] for (ϵ,δ)-differential privacy can be loose. Hence, we use RDP, which offers a much simpler composition rule that is shown to be tight. Here, we review the properties of RDP [34] that we utilize in our analysis in Section 3.

**Proposition** **1**(From RDP to Differential Privacy [34])**.**
*If A is an $\left(\alpha ,\epsilon_{r}\right)$-RDP mechanism, then it also satisfies $\left(\epsilon_{r}+\frac{\log\frac{1}{\delta_{r}}}{\alpha -1},\delta_{r}\right)$-differential privacy for any $0<\delta_{r}<1$.*

**Proposition** **2**(Composition of RDP [34])**.**
*Let $A:D\mapsto T_{1}$ be $\left(\alpha ,\epsilon_{r1}\right)$-RDP and $B:D\mapsto T_{2}$ be $\left(\alpha ,\epsilon_{r2}\right)$-RDP. Then the mechanism defined as (X,Y), where X∼A(D) and Y∼B(X,D), satisfies $\left(\alpha ,\epsilon_{r1}+\epsilon_{r2}\right)$-RDP.*

**Proposition** **3**(RDP and Gaussian Mechanism [34])**.**
*If A has $L_{2}$-sensitivity 1, then the Gaussian mechanism $G_{\sigma }A\left(D\right)=A\left(D\right)+E$, where $E∼N(0,\sigma^{2})$ satisfies $\left(\alpha ,\frac{\alpha }{2\sigma^{2}}\right)$-RDP. Additionally, a composition of T Gaussian mechanisms satisfies $\left(\alpha ,\frac{\alpha T}{2\sigma^{2}}\right)$-RDP.*

**Correlation Assisted Private Estimation (CAPE) [6].** As mentioned before, we utilize the CAPE protocol for developing capeFM. In Section 5.2 we describe the CAPE trust/collusion model in detail, and discuss how the correlated noise in a decentralized-data setting is used to reduce the excess noise introduced in conventional decentralized DP algorithms. We use the terms “distributed” and “decentralized” interchangeably in this paper. Note that the CAPE scheme, and consequently the proposed capeFM algorithm can be readily extended (see Section III.C of Imtiaz et al. [6]) for federated learning [29] settings.

The CAPE protocol considers a decentralized data setting with *S* sites and a central aggregator node in an “honest but curious” threat model [6]. For simplicity, we consider the symmetric setting: each site s∈[S] holds a dataset of $N_{s}=\frac{N}{S}$ disjoint data samples, where the total number of samples across all sites is *N*. CAPE overcomes the utility degradation in conventional decentralized DP schemes and achieves the same noise variance as that of the pooled-data scenario in certain parameter regimes. The privacy of CAPE is given by Theorem 1 and the claim that the noise variance of the estimator is exactly the same as if all data were present at the aggregator is formalized in Lemma 1. Here, we review the relevant properties of the CAPE scheme for extending our proposed Gaussian FM to the decentralized-data setting. We refer the reader to Imtiaz et al. [6] for the proofs of these properties.

**Theorem** **1**(Privacy of CAPE scheme [6])**.**
*In a decentralized data setting with $N_{s}=\frac{N}{S}$ and $\tau_{s}^{2}=\tau^{2}$ for all sites s∈[S], if at most $S_{C}=\left\lceil \frac{S}{3}\right\rceil -1$ collude after execution, then CAPE guarantees (ϵ,δ)-differential privacy for each site, where (ϵ,δ) satisfy the relation $\delta =2\frac{\sigma_{z}}{\epsilon -\mu_{z}}\phi \left(\frac{\epsilon -\mu_{z}}{\sigma_{z}}\right)$, ϵ∈(0,1) and $\left(\mu_{z},\sigma_{z}\right)$ are given by*
$$\begin{array}{cc} \mu_{z} & =\frac{S^{3}}{2\tau^{2}N^{2}\left(1+S\right)}\left(\frac{S-S_{C}+2}{S-S_{C}}+\frac{\frac{9}{S-S_{C}}S_{C}^{2}}{S\left(1+S\right)-3S_{C}^{2}}\right),\\ \sigma_{z} & =2\mu_{z}. \end{array}$$

**Lemma** **1**([6])**.**
*Consider the symmetric setting: $N_{s}=\frac{N}{S}$ and $\tau_{s}^{2}=\tau^{2}$ for all sites s∈[S]. Let the variances of the noise terms $e_{s}$ and $g_{s}$ be $\tau_{e}^{2}=\left(1-\frac{1}{S}\right)\tau_{s}^{2}$ and $\tau_{g}^{2}=\frac{\tau_{s}^{2}}{S}$, respectively. If we denote the variance of the additive noise (for preserving privacy) in the pooled-data scenario by $\tau_{pool}^{2}$ and the variance of the estimator $a_{cape}$ by $\tau_{cape}^{2}$ then CAPE protocol achieves the same noise variance as the pooled-data scenario (i.e., $\tau_{pool}^{2}=\tau_{cape}^{2}$).*

**Proposition** **4**(Performance improvement using CAPE [6])**.**
*If the local noise variances are $\left\{\tau_{s}^{2}\right\}$ for s∈[S] then the CAPE scheme provides a reduction $G=\frac{\tau_{conv}^{2}}{\tau_{cape}^{2}}=S$ in noise variance over the conventional decentralized DP scheme in the symmetric setting ($N_{s}=\frac{N}{S}$ and $\tau_{s}^{2}=\tau^{2}$∀s∈[S]), where $\tau_{conv}^{2}$ and $\tau_{cape}^{2}$ are the noise variances of the final estimate at the aggregator in the conventional scheme and the CAPE scheme, respectively.*

**Proposition** **5**(Scope of CAPE [6])**.**
*Consider a decentralized setting with S>1 sites in which site s∈[S] has a dataset $D_{s}$ of $N_{s}$ samples and $\sum_{s=1}^{S}N_{s}=N$. Suppose the sites are employing the CAPE scheme to compute a function f(D) with $L_{2}$-sensitivity Δ(N). Denote $n=[N_{1},N_{2},\ldots ,N_{S}]$ and observe the ratio $H\left(n\right)=\frac{\tau_{cape}^{2}}{\tau_{pool}^{2}}=\frac{\sum_{s=1}^{S}\Delta^{2}\left(N_{s}\right)}{S^{3}\Delta^{2}\left(N\right)}$. Then the CAPE protocol achieves H(n)=1, if (i) $\Delta \left(\frac{N}{S}\right)=S\Delta \left(N\right)$ for convex Δ(N); and (ii) $S^{3}\Delta^{2}\left(N\right)=\sum_{s=1}^{S}\Delta^{2}\left(N_{s}\right)$ for general Δ(N).*

### 2.2. Functional Mechanism [7]

In this section, we first review the existing functional mechanism through a regression model following [7] before describing our proposed improvement. Let D be a dataset that contains *N* samples of the form $\left(x_{n},y_{n}\right)$, where $x_{n}\in R^{D}$ is the feature vector and $y_{n}\in R$ is the response for n∈[N]. Without loss of generality, we assume for each sample that $∥x_{n}{∥}_{2}\leq 1$. The objective is to construct a regression model that enables one to predict any $y_{n}$ based on $x_{n}$. Depending on the regression model, the mapping function can be of various types. Without loss of generality, it can be parameterized with a *D*-dimensional vector w of real numbers. To evaluate whether w leads to an accurate model, a *cost function f* is defined to measure the deviation between the original and predicted values of $y_{n}$, given w as the model parameters. The optimal model parameter $w^{*}$ is defined as
$$w^{*}=\arg\min_{w}f_{D}\left(w\right),$$
where the empirical average cost function is
(1)$$f_{D}\left(w\right)=\frac{1}{N}\sum_{n=1}^{N}f\left(x_{n},w\right).$$
Note that $f_{D}\left(w\right)$ depends on the data samples. In cases where the data are privacy-sensitive, the empirical average cost function $f_{D}\left(w\right)$ (or any function computed from it, such as its gradient or the optimizer $w^{*}$) may reveal private information about the members of the dataset. To make the model differentially private, one approach is to add noise to the gradients of the cost function at every iteration [12]. We refer to this approach as *noisy gradient descent* in this paper. Another approach is the to perturb the objective function [7,8,9,10]. In particular, the original FM [7] and the relaxed FM [10] use a randomized approximation of the objective function.

Now, recall that $w\in R^{D}$ contains the model parameters $w={\left[w_{1},w_{2},\ldots ,w_{D}\right]}^{⊤}$. We define $\phi \left(w\right)=w_{1}^{c_{1}}w_{2}^{c_{2}}\ldots w_{D}^{c_{D}}$ for some $c_{1},c_{2},\ldots ,c_{D}\in N$. Let $\Phi_{j}$ denote the set of all ϕ(w) with degree j∈N, i.e.,
$$\Phi_{j}=\left\{w_{1}^{c_{1}}w_{2}^{c_{2}}\ldots w_{D}^{c_{D}}|\sum_{d=1}^{D}c_{d}=j\right\}.$$
For example, $\Phi_{0}=\left\{1\right\}$, $\Phi_{1}=\left\{w_{1},w_{2},\ldots ,w_{D}\right\}$, and $\Phi_{2}=\left\{w_{d_{1}}w_{d_{2}}|d_{1},d_{2}\in \left[D\right]\right\}$. By the Stone–Weierstrass Theorem [35], any continuous and differentiable $f(x_{n},w)$ can be *always* written as a (potentially infinite) sum of monomials of $\left\{w_{d}\right\}$, i.e., for some J∈[0,∞), we have
$$f\left(x_{n},w\right)=\sum_{j=0}^{J}\sum_{\phi \in \Phi_{j}}\lambda_{\phi n}\phi \left(w\right),$$
where $\lambda_{\phi n}\in R$ denotes the coefficient of ϕ(w) in the polynomial. Note that $\lambda_{\phi n}$ is a function of the *n*-th data sample. Consequently, the $f(x_{n},w)$ as expressed above depends on the model parameters through ϕ(w) and on the data samples through $\lambda_{\phi n}$. The expression for average cost in (Equation 1) can now be written as
(2)$$f_{D}\left(w\right)=\frac{1}{N}\sum_{n=1}^{N}\sum_{j=0}^{J}\sum_{\phi \in \Phi_{j}}\lambda_{\phi n}\phi \left(w\right)=\sum_{j=0}^{J}\sum_{\phi \in \Phi_{j}}\left(\frac{1}{N}\sum_{n=1}^{N}\lambda_{\phi n}\right)\phi \left(w\right).$$
For regression analysis on two neighboring datasets D and $D^{\prime }$ differing in a single sample, the $L_{1}$-sensitivity of the data-dependent term in (Equation 2) is computed as [7]:$$\sum_{j=0}^{J}\sum_{\phi \in \Phi_{j}}\frac{1}{N}∥\left(\sum_{D}\lambda_{\phi n}-\sum_{D^{\prime }}\lambda_{\phi n}\right){∥}_{1}\leq \frac{2}{N}\max_{n}\sum_{j=0}^{J}\sum_{\phi \in \Phi_{j}}∥\lambda_{\phi n}{∥}_{1}≜\Delta^{fm}.$$
In FM, Zhang et al. [7] proposed to perturb $f_{D}\left(w\right)$ by injecting Laplace noise with variance $2{\left(\frac{\Delta^{fm}}{\epsilon }\right)}^{2}$ into each coefficient of the polynomial. FM achieves ϵ-DP by obtaining the optimal model parameters ${\hat{w}}^{*}$ that minimize the noise-perturbed function ${\hat{f}}_{D}\left(w\right)$.

As mentioned before, decomposition such as (Equation 2) can be performed for any continuous and differentiable cost function $f(x_{n},w)$. However, depending on the complexity of $f(x_{n},w)$, the decomposition may be non-trivial. In Section 4, we show how such decomposition can be performed on linear regression and logistic regression problems, as illustrative examples.

## 3. Functional Mechanism with Approximate Differential Privacy: Gaussian FM

Zhang et al. [7] computed the $L_{1}$-sensitivity $\Delta^{fm}$ of the data-dependent terms for linear regression and logistic regression problems. The $\Delta^{fm}$ is shown to be $\frac{2}{N}{\left(1+D\right)}^{2}$ for linear regression, and $\frac{1}{N}\left(\frac{D^{2}}{4}+3D\right)$ for logistic regression. We note that $\Delta^{fm}$ grows quadratically with the ambient dimension of the data samples, resulting in a excessively large amount of noise to be injected into the objective function. Additionally, Ding et al. [10] proposed relaxed FM, a “utility-enhancement scheme”, by replacing the Laplace mechanism with the Extended Gaussian mechanism [11], and thus achieving slightly better utility than the original FM at the expense of an approximate DP guarantee instead of a pure DP guarantee. However, Ding et al. [10] showed that the $L_{2}$-sensitivity of the data-dependent terms for the logistic regression problem is $\Delta^{rlx-fm}=\frac{1}{N}\sqrt{\frac{D^{2}}{16}+D}$. Additionally, using the technique outlined in [10], it can be shown that the $L_{2}$-sensitivity of the data-dependent terms is $\Delta^{rlx-fm}=\frac{2}{N}\sqrt{1+4D+D^{2}}$ for the linear regression problem (please see Appendix A for details). For both cases, we observe that $\Delta^{rlx-fm}$ grows linearly with the ambient dimension of the data samples. Therefore, the privacy-preserving additive noise variances in both the original FM and relaxed FM schemes are data-dimensionality dependent, and therefore, can be prohibitively large even for moderate *D*. Moreover, both FM and relaxed FM schemes add the same amount of noise to each polynomial coefficient $\lambda_{\phi n}$ irrespective of the order *j*. With a tighter characterization, we show in Section 4 that the sensitivities of these coefficients are different for different order *j*. We reduce the amount of added noise by addressing these issues and performing a novel sensitivity analysis. The key points are as follows:Instead of computing the ϵ-DP approximation of the objective function using the Laplace mechanism, we use the Gaussian mechanism to compute the (ϵ,δ)-DP approximation of $f_{D}\left(w\right)$. This gives a weaker privacy guarantee than the *pure* differential privacy, but provides much better *utility*.Recall that the original FM achieves ϵ-DP by adding Laplace noise scaled to the $L_{1}$-sensitivity of the data-dependent terms of the objective function $f_{D}\left(w\right)$ in (Equation 2). As we use the Gaussian mechanism, we require $L_{2}$-sensitivity analysis. To compute the $L_{2}$-sensitivity of the data-dependent terms of the objective function $f_{D}\left(w\right)$ in (Equation 2), we first define an *array* $\Lambda_{j}$ that contains $\frac{1}{N}\sum_{n=1}^{N}\lambda_{\phi n}$ as its entries for all $\phi \left(w\right)\in \Phi_{j}$. The term “array” is used because the dimension of $\Lambda_{j}$ depends on the cardinality of $\Phi_{j}$. For example, for j=0, $\Lambda_{0}$ is a scalar because $\Phi_{0}=\left\{1\right\}$; for j=1, $\Lambda_{1}$ can be expressed as a *D*-dimensional vector because $\Phi_{1}=\left\{w_{1},w_{2},...,w_{D}\right\}$; for j=2, $\Lambda_{2}$ can be expressed as a D×D matrix because $\Phi_{2}=\left\{w_{d_{1}}w_{d_{2}}|d_{1},d_{2}\in \left[D\right]\right\}$.

We rewrite the objective function as
(3)$$f_{D}\left(w\right)=\sum_{j=0}^{J}\sum_{\phi \in \Phi_{j}}\left(\frac{1}{N}\sum_{n=1}^{N}\lambda_{\phi n}\right)\phi \left(w\right)=\sum_{j=0}^{J}\langle \Lambda_{j},{\bar{\phi }}_{j}\rangle ,$$
where ${\bar{\phi }}_{j}$ is the array containing all $\phi \left(w\right)\in \Phi_{j}$ as its entries. Note that ${\bar{\phi }}_{j}$ and $\Lambda_{j}$ have the same dimensions and number of elements. We define the $L_{2}$-sensitivity of $\Lambda_{j}$ as
(4)$$\Delta_{j}=\max_{D,D^{\prime }}∥\Lambda_{j}^{D}-\Lambda_{j}^{D^{\prime }}{∥}_{2},$$
where $\Lambda_{j}^{D}$ and $\Lambda_{j}^{D^{\prime }}$ are computed on neighboring datasets D and $D^{\prime }$, respectively. Following the Gaussian mechanism [33], we can calculate the (ϵ,δ) differentially private estimate of $\Lambda_{j}$, denoted ${\hat{\Lambda }}_{j}$ as
(5)$${\hat{\Lambda }}_{j}=\Lambda_{j}+e_{j},$$
where the noise array $e_{j}$ has the same dimension as $\Lambda_{j}$, and contains entries drawn i.i.d. from $N(0,\tau_{j}^{2})$ with $\tau_{j}=\frac{\Delta_{j}}{\epsilon }\sqrt{2\log\frac{1.25}{\delta }}$. Finally, we have
(6)$${\hat{f}}_{D}\left(w\right)=\sum_{j=0}^{J}\langle {\hat{\Lambda }}_{j},{\bar{\phi }}_{j}\rangle .$$

As the function $f_{D}\left(w\right)$ depends on the data only through $\left\{\Lambda_{j}\right\}$, this computation satisfies (ϵ,δ)-differential privacy. Our proposed Gaussian FM is shown in detail in Algorithm 1.

**Theorem** **2**(Privacy of the Gaussian FM (Algorithm 1))**.**
*Consider Algorithm 1 with privacy parameters (ϵ,δ), and the empirical average cost function $f_{D}\left(w\right)$ represented as in (Equation 3). Then Algorithm 1 computes an (ϵ,δ) differentially private approximation ${\hat{f}}_{D}\left(w\right)$ to $f_{D}\left(w\right)$. Consequently, the minimizer ${\hat{w}}^{*}={arg min}_{w}{\hat{f}}_{D}\left(w\right)$ satisfies (ϵ,δ)-differential privacy.*

**Algorithm 1** Gaussian FM
**Require:** Data samples $\left(x_{n},y_{n}\right)$ for n∈[N]; cost function $f_{D}\left(w\right)$ represented as in (Equation 3); privacy parameters (ϵ,δ).1:**for** 
0≤j≤J 
**do**2:    Compute $\Lambda_{j}$ as shown in Section 43:    Compute $\Delta_{j}=\max_{D,D^{\prime }}∥\Lambda_{j}^{D}-\Lambda_{j}^{D^{\prime }}{∥}_{2}$4:    Compute $\tau_{j}=\frac{\Delta_{j}}{\epsilon }\sqrt{2\log\frac{1.25}{\delta }}$5:    Compute $e_{j}∼N\left(0,\tau_{j}^{2}\right)$ with the same dimension as $\Lambda_{j}$6:    Release ${\hat{\Lambda }}_{j}=\Lambda_{j}+e_{j}$7:
**end for**
8:Compute ${\hat{f}}_{D}\left(w\right)=\sum_{j=0}^{J}\langle {\hat{\Lambda }}_{j},{\bar{\phi }}_{j}\rangle$9:**return** Perturbed objective function ${\hat{f}}_{D}\left(w\right)$


**Proof.** The proof of Theorem 2 follows from the fact that the function ${\hat{f}}_{D}\left(w\right)$ depends on the data samples only through $\left\{{\hat{\Lambda }}_{j}\right\}$. The computation of $\left\{{\hat{\Lambda }}_{j}\right\}$ is (ϵ,δ)-differentially private by the Gaussian mechanism [4,33]. Therefore, the release of ${\hat{f}}_{D}\left(w\right)$ satisfies (ϵ,δ)-differential privacy. One way to rationalize this is to consider that the probability of the event of selecting a particular set of $\left\{{\hat{\Lambda }}_{j}\right\}$ is the same as the event of formulating a function ${\hat{f}}_{D}\left(w\right)$ with that set of $\left\{{\hat{\Lambda }}_{j}\right\}$. Therefore, it suffices to consider the joint density of the $\left\{{\hat{\Lambda }}_{j}\right\}$ and find an upper bound on the ratio of the joint densities of the $\left\{{\hat{\Lambda }}_{j}\right\}$ under two neighboring datasets D and $D^{\prime }$. As we employ the Gaussian mechanism to compute $\left\{{\hat{\Lambda }}_{j}\right\}$, the ratio is upper bounded by exp(ϵ) with probability at least 1−δ. Therefore, the release of ${\hat{f}}_{D}\left(w\right)$ satisfies (ϵ,δ)-differential privacy. Furthermore, differential privacy is post-processing invariant. Therefore, the computation of the minimizer ${\hat{w}}^{*}={arg min}_{w}{\hat{f}}_{D}\left(w\right)$ also satisfies (ϵ,δ)-differential privacy.    □

**Privacy Analysis of Noisy Gradient Descent [12] using Rényi Differential Privacy.** One of the most crucial qualitative properties of DP is that it allows us to evaluate the cumulative privacy loss over multiple computations [33]. Cumulative, or total, privacy loss is different from (ϵ,δ)-DP in multi-round machine learning algorithms. In order to demonstrate the superior privacy guarantee of the proposed Gaussian FM, we compare it to the existing functional mechanism [7], the relaxed functional mechanism [10], the objective perturbation [8], and the noisy gradient descent [12] method. Note that, similar to objective perturbation, FM and relaxed FM, the proposed Gaussian FM injects randomness in a single round, and therefore does not require privacy accounting. However, the noisy gradient descent method involves addition of noise in each step the gradient is computed. That is, noise is added to the computed gradients of the parameters of the objective function during optimization. Since it is a multi-round algorithm, the overall ϵ used during optimization is different from the ϵ for every iteration. We follow the analysis procedure outlined in [6] for the privacy accounting of the noisy gradient descent algorithm. Note that Proposition 3 described in Section 2.1 is defined for functions with unit $L_{2}$-sensitivity. Therefore, if a noise from $N(0,\tau^{2})$ is added to a function with sensitivity Δ, then the resulting mechanism satisfies $\left(\alpha ,\frac{\alpha }{2\frac{\tau^{2}}{\Delta^{2}}}\right)$-RDP. Now, according to Proposition 3, the *T*-fold composition of Gaussian mechanisms satisfies $\left(\alpha ,\frac{\alpha T}{2\frac{\tau^{2}}{\Delta^{2}}}\right)$-RDP. Finally, according to Proposition 1, it also satisfies $\left(\epsilon_{r}+\frac{\log\frac{1}{\delta_{r}}}{\alpha -1},\delta_{r}\right)$-differential privacy for any $0\leq \delta_{r}\leq 1$, where $\epsilon_{r}=\frac{\alpha T}{2\frac{\tau^{2}}{\Delta^{2}}}$. For a given value of $\delta_{r}$, we can express the value of the optimal overall $\epsilon_{\mathrm{opt}}$ as a function of $\alpha_{\mathrm{opt}}$:(7)$$\epsilon_{\mathrm{opt}}=\frac{\alpha_{\mathrm{opt}}T}{2\frac{\tau^{2}}{\Delta^{2}}}+\frac{\log\frac{1}{\delta_{r}}}{\alpha_{\mathrm{opt}}-1},$$
where $\alpha_{\mathrm{opt}}$ is given by
(8)$$\alpha_{\mathrm{opt}}=1+\sqrt{\frac{2}{T}\frac{\tau^{2}}{\Delta^{2}}\log\frac{1}{\delta_{r}}}.$$
We compute the overall ϵ following this procedure for the noisy gradient descent algorithm [12] in our experiments in Section 6.

## 4. Application of Gaussian FM in Regression Analysis

In this section, we demonstrate how our proposed Gaussian FM can be applied to linear and logistic regression problems to achieve (ϵ,δ)-DP. For both cases, we first decompose the objective function (i.e., the empirical average cost function) into a finite series of polynomials, inject noise into the coefficients (i.e., the only data-dependent components in the decomposition) using Gaussian mechanism, and finally minimize the (ϵ,δ)-differentially private objective function. As before, we assume that we have a dataset D with *N* samples of the form $\left(x_{n},y_{n}\right)$, where for each sample n∈[N], the *D*-dimensional feature vector is $x_{n}={\left[x_{n1}\;x_{n2}\;\cdots \;x_{nD}\right]}^{⊤}$ (normalized to ensure $∥x_{n}{∥}_{2}\leq 1$) and the corresponding output is $y_{n}$.

### 4.1. Linear Regression

For our linear regression problem, we assume $y_{n}\in \left[-1,1\right]$. Let $w\in R^{D}$ be the parameter vector. The goal of linear regression is to find the optimal $w^{*}$ so that $x_{n}^{⊤}w^{*}\approx y_{n}$. The empirical average cost function is defined as
(9)$$f_{D}\left(w\right)=\frac{1}{N}\sum_{n=1}^{N}{\left(y_{n}-x_{n}^{⊤}w\right)}^{2}.$$
Using simple algebra, this equation can be decomposed into a series of polynomials as
$$f_{D}\left(w\right)=\left(\frac{1}{N}\sum_{n=1}^{N}y_{n}^{2}\right)+\sum_{d=1}^{D}\left(-\frac{2}{N}\sum_{n=1}^{N}y_{n}x_{nd}\right)w_{d}+\sum_{d_{1}=1}^{D}\sum_{d_{2}=1}^{D}\left(\frac{1}{N}\sum_{n=1}^{N}x_{nd_{1}}x_{nd_{2}}\right)w_{d_{1}}w_{d_{2}}.$$
As we intend to compute the differentially private minimizer ${\hat{w}}^{*}$, we observe that the representation of $f_{D}\left(w\right)$ is of the form $f_{D}\left(w\right)=\sum_{j=0}^{J}\langle \Lambda_{j},{\bar{\phi }}_{j}\rangle$ with J=2. The expressions for $\Lambda_{j}$ are
$$\begin{array}{cc} \Lambda_{0} & =\frac{1}{N}\sum_{n=1}^{N}y_{n}^{2},\\ \Lambda_{1} & =-\frac{2}{N}\left[\begin{array}{c} \sum_{n=1}^{N}y_{n}x_{n1}\\ \sum_{n=1}^{N}y_{n}x_{n2}\\ \vdots \\ \sum_{n=1}^{N}y_{n}x_{nD} \end{array}\right],\\ \Lambda_{2} & =\frac{1}{N}\left[\begin{array}{ccc} \sum_{n=1}^{N}x_{n1}^{2} & \cdots & \sum_{n=1}^{N}x_{n1}x_{nD}\\ \vdots & \ddots & \vdots \\ \sum_{n=1}^{N}x_{nD}x_{n1} & \cdots & \sum_{n=1}^{N}x_{nD}^{2} \end{array}\right]=\frac{1}{N}\left(XX^{⊤}\right). \end{array}$$
Here, $\Lambda_{0}$ is a scalar, $\Lambda_{1}$ is a *D*-dimensional vector, and $\Lambda_{2}$ is a D×D symmetric matrix, since X is an D×N matrix containing $x_{n}$ as its columns. The expressions for ${\bar{\phi }}_{j}$ are
$$\begin{array}{cc} {\bar{\phi }}_{0} & =1,\\ {\bar{\phi }}_{1} & =\left[\begin{array}{c} w_{1}\\ w_{2}\\ \vdots \\ w_{D} \end{array}\right],\\ {\bar{\phi }}_{2} & =\left[\begin{array}{cccc} w_{1}^{2} & w_{1}w_{2} & \cdots & w_{1}w_{D}\\ w_{2}w_{1} & w_{2}^{2} & \cdots & w_{2}w_{D}\\ \vdots & \vdots & \ddots & \vdots \\ w_{D}w_{1} & w_{D}w_{2} & \cdots & w_{D}^{2} \end{array}\right]. \end{array}$$
The next step is finding the sensitivities of $\left\{\Lambda_{j}\right\}$ using (Equation 4). Let D and $D^{\prime }$ be two neighboring datasets differing in only one sample, e.g., the last samples $\left(x_{N},y_{N}\right)$ and $\left(x_{N}^{\prime },y_{N}^{\prime }\right)$. Now, the $L_{2}$-sensitivity of $\Lambda_{0}$ is
$$\begin{array}{cc} \Delta_{0} & =\max_{D,D^{\prime }}∥\frac{1}{N}\sum_{n=1}^{N}y_{n}^{2}-\frac{1}{N}\sum_{n=1}^{N}y_{n}^{\prime 2}{∥}_{2}\\ \; & =\frac{1}{N}\max_{D,D^{\prime }}∥y_{N}^{2}-y_{N}^{\prime 2}{∥}_{2}\\ \; & \leq \frac{1}{N}, \end{array}$$
since $y_{n}\in \left[-1,1\right]$ and hence $y_{n}^{2}\in \left[0,1\right]$. Next, the $L_{2}$-sensitivity of $\Lambda_{1}$ is
$$\begin{array}{cc} \Delta_{1} & =\max_{D,D^{\prime }}∥-\frac{2}{N}y_{N}x_{N}+\frac{2}{N}y_{N}^{\prime }x_{N}^{\prime }{∥}_{2}\\ \; & \leq \frac{2}{N}\max_{D,D^{\prime }}\left(∥y_{N}x_{N}{∥}_{2}+∥y_{N}^{\prime }x_{N}^{\prime }{∥}_{2}\right)\\ \; & =\frac{2}{N}\max_{D,D^{\prime }}\left(|y_{N}|∥x_{N}{∥}_{2}+|y_{N}^{\prime }|∥x_{N}^{\prime }{∥}_{2}\right)\\ \; & \leq \frac{4}{N}, \end{array}$$
where the second line follows from the triangle inequality, and the last line follows from the assumptions that $y_{n}\in \left[-1,1\right]$ and $∥x_{n}{∥}_{2}\leq 1$. Finally, the $L_{2}$-sensitivity of $\Lambda_{2}$ is
$$\begin{array}{cc} \Delta_{2} & =\max_{D,D^{\prime }}∥\frac{1}{N}\left(XX^{⊤}\right)-\frac{1}{N}\left(X^{\prime }X^{\prime ⊤}\right){∥}_{2}\\ \; & =\frac{1}{N}\max_{D,D^{\prime }}∥x_{N}x_{N}^{⊤}-x_{N}^{\prime }x_{N}^{\prime ⊤}{∥}_{2}\\ \; & \leq \frac{1}{N}. \end{array}$$
The proof of the inequality in the last line is as follows:

**Proof.** The term $\left(x_{N}x_{N}^{⊤}-x_{N}^{\prime }x_{N}^{\prime ⊤}\right)$ is a D×D symmetric matrix, whose norm can be expressed [41] as $\sup\left\{u^{⊤}\left[x_{N}x_{N}^{⊤}-x_{N}^{\prime }x_{N}^{\prime ⊤}\right]v\;|\;u=v,\;∥u{∥}_{2}=∥v{∥}_{2}=1\right\}$. It follows that
$$\begin{array}{cc} ∥x_{N}x_{N}^{⊤}-x_{N}^{\prime }x_{N}^{\prime ⊤}{∥}_{2} & =\sup\left\{u^{⊤}x_{N}x_{N}^{⊤}u-u^{⊤}x_{N}^{\prime }x_{N}^{\prime ⊤}u\right\}\\ \; & =\sup\left\{{\left(x_{N}^{⊤}u\right)}^{⊤}\left(x_{N}^{⊤}u\right)-{\left(x_{N}^{\prime ⊤}u\right)}^{⊤}\left(x_{N}^{\prime ⊤}u\right)\right\}\\ \; & =\sup\left\{∥x_{N}^{⊤}u{∥}_{2}^{2}-∥x_{N}^{\prime ⊤}u{∥}_{2}^{2}\right\}\\ \; & \leq \sup\left\{∥x_{N}^{⊤}{∥}_{2}^{2}∥u{∥}_{2}^{2}-∥x_{N}^{\prime ⊤}{∥}_{2}^{2}∥u{∥}_{2}^{2}\right\}\leq 1. \end{array}$$   □

After computing the $L_{2}$-sensitivity of $\Lambda_{j}$ for j=0,1, and 2, we can now compute the noise array $e_{j}∼N\left(0,\tau_{j}^{2}\right)$, where $\tau_{j}=\frac{\Delta_{j}}{\epsilon }\sqrt{2\log\frac{1.25}{\delta }}$, and then compute $\left\{{\hat{\Lambda }}_{j}\right\}$ following (Equation 5). Using these, we can compute the (ϵ,δ) differentially private ${\hat{f}}_{D}\left(w\right)$ according to (Equation 6), and consequently, the minimizer ${\hat{w}}^{*}={arg min}_{w}{\hat{f}}_{D}\left(w\right)$. Note that, unlike the existing FM and relaxed FM, the additive noise variances of our proposed Gaussian FM do not depend on the sample dimension *D*. More specifically, for the linear regression problem, the $L_{1}$-sensitivity of the coefficients in FM [7] is $\Delta^{fm}=\frac{2}{N}{\left(1+D\right)}^{2}$ and the $L_{2}$-sensitivity of the coefficients in relaxed FM [10] is $\Delta^{rlx-fm}=\frac{2}{N}\sqrt{1+4D+D^{2}}$ (see Appendix A for the proof). Both of these sensitivities are orders of magnitude larger than $\Delta_{j}$ that we achieved for j∈{0,1,2}, and for practical values of *D* and *N*. Thus, the proposed Gaussian FM can offer the (ϵ,δ)-differentially private approximation ${\hat{f}}_{D}\left(w\right)$ with much less noise, which results in a (ϵ,δ)-differentially private model ${\hat{w}}^{*}$ that is much closer to the true model $w^{*}$. We show empirical validation on synthetic and real datasets in Section 6.

### 4.2. Logistic Regression

For the logistic regression problem, we assume $y_{n}\in \left\{0,1\right\}$ to be the class labels. The class label is approximated using the *sigmoid* function defined as $f_{sig}\left(z\right)=\frac{1}{1+\exp(-z)}$. Let $w\in R^{D}$ be the parameter vector. The goal of logistic regression is to find the optimal $w^{*}$ so that $f_{sig}\left(x_{n}^{⊤}w^{*}\right)\approx y_{n}$. The empirical average cost function for logistic regression is defined as
(10)$$\begin{array}{cc} f_{D}\left(w\right) & =-\frac{1}{N}\sum_{n=1}^{N}y_{n}\log\left(f_{sig}\left(x_{n}^{⊤}w\right)\right)+\left(1-y_{n}\right)\log\left(1-f_{sig}\left(x_{n}^{⊤}w\right)\right)\\ \; & =\frac{1}{N}\sum_{n=1}^{N}\log\left(1+\exp(x_{n}^{⊤}w)\right)-y_{n}x_{n}^{⊤}w. \end{array}$$
Unlike linear regression, the simplified form of $f_{D}\left(w\right)$ in the second line cannot be represented with a finite series of polynomials. Zhang et al. [7] proposed an *approximate* polynomial form of $f_{D}\left(w\right)$ using Taylor series expansion, written as
$${\tilde{f}}_{D}\left(w\right)=\frac{1}{N}\sum_{n=1}^{N}\sum_{k=0}^{2}\frac{f_{1}^{\left(k\right)}\left(0\right)}{k!}{\left(x_{n}^{⊤}w\right)}^{k}-\frac{1}{N}\sum_{n=1}^{N}y_{n}x_{n}^{⊤}w.$$
Using simple algebra and the values of $f_{1}^{\left(k\right)}\left(0\right)$ for k=0,1, and 2, i.e., $f_{1}^{\left(0\right)}\left(0\right)=\log2$, $f_{1}^{\left(1\right)}\left(0\right)=\frac{1}{2}$, and $f_{1}^{\left(k\right)}\left(0\right)=\frac{1}{4}$, we obtain
$${\tilde{f}}_{D}\left(w\right)=\log2+\sum_{d=1}^{D}\left(\frac{1}{N}\sum_{n=1}^{N}\left(\frac{1}{2}-y_{n}\right)x_{nd}\right)w_{d}+\sum_{d_{1}=1}^{D}\sum_{d_{2}=1}^{D}\left(\frac{1}{8N}\sum_{n=1}^{N}x_{nd_{1}}x_{nd_{2}}\right)w_{d_{1}}w_{d_{2}}.$$
As before, we intend to compute the differentially private minimizer ${\hat{w}}^{*}$, and we observe that the representation of ${\tilde{f}}_{D}\left(w\right)$ is of the form $f_{D}\left(w\right)=\sum_{j=0}^{J}\langle \Lambda_{j},{\bar{\phi }}_{j}\rangle$ with J=2. The expressions for $\Lambda_{j}$ are
$$\begin{array}{cc} \Lambda_{0} & =\log2,\\ \Lambda_{1} & =\frac{1}{N}\left[\begin{array}{c} \sum_{n=1}^{N}\left(\frac{1}{2}-y_{n}\right)x_{n1}\\ \sum_{n=1}^{N}\left(\frac{1}{2}-y_{n}\right)x_{n2}\\ \vdots \\ \sum_{n=1}^{N}\left(\frac{1}{2}-y_{n}\right)x_{nD} \end{array}\right],\\ \Lambda_{2} & =\frac{1}{8N}\left[\begin{array}{ccc} \sum_{n=1}^{N}x_{n1}^{2} & \cdots & \sum_{n=1}^{N}x_{n1}x_{nD}\\ \vdots & \ddots & \vdots \\ \sum_{n=1}^{N}x_{nD}x_{n1} & \cdots & \sum_{n=1}^{N}x_{nD}^{2} \end{array}\right]=\frac{1}{8N}\left(XX^{⊤}\right). \end{array}$$
Again, $\Lambda_{j}$ is a scalar, a *D*-dimensional vector, and a D×D matrix for j=0,1, and 2, respectively. We can express ${\bar{\phi }}_{j}$ for j=0,1, and 2 the same way as we did for linear regression in Section 4.1. To compute the sensitivities of $\left\{\Lambda_{j}\right\}$ using (Equation 4), let D and $D^{\prime }$ be two neighboring datasets differing in only the last samples, which are $\left(x_{N},y_{N}\right)$ and $\left(x_{N}^{\prime },y_{N}^{\prime }\right)$, respectively. Now, the $L_{2}$-sensitivity of $\Lambda_{0}$ is $\Delta_{0}=\max_{D,D^{\prime }}∥\log2-\log2{∥}_{2}=0$. The $L_{2}$-sensitivity of $\Lambda_{1}$ is
$$\begin{array}{cc} \Delta_{1} & =\max_{D,D^{\prime }}∥\frac{1}{N}\left(\frac{1}{2}-y_{N}\right)x_{N}-\frac{1}{N}\left(\frac{1}{2}-y_{N}^{\prime }\right)x_{N}^{\prime }{∥}_{2}\\ \; & \leq \frac{1}{N}\max_{D,D^{\prime }}\left(|\frac{1}{2}-y_{N}|∥x_{N}{∥}_{2}+|\frac{1}{2}-y_{N}^{\prime }|∥x_{N}^{\prime }{∥}_{2}\right)\\ \; & \leq \frac{1}{N}, \end{array}$$
where $|\frac{1}{2}-y_{N}|\leq \frac{1}{2}$, since $y_{n}\in \left\{0,1\right\}$, and $∥x_{n}{∥}_{2}\leq 1$. Finally, the $L_{2}$-sensitivity of $\Lambda_{2}$ is
$$\begin{array}{cc} \Delta_{2} & =\max_{D,D^{\prime }}∥\frac{1}{8N}\left(XX^{⊤}\right)-\frac{1}{8N}\left(X^{\prime }X^{\prime ⊤}\right){∥}_{2}\\ \; & =\frac{1}{8N}\max_{D,D^{\prime }}∥x_{N}x_{N}^{⊤}-x_{N}^{\prime }x_{N}^{\prime ⊤}{∥}_{2}\\ \; & \leq \frac{1}{8N}, \end{array}$$
where the inequality follows from the expression for the norm of a symmetric matrix, as shown in Section 4.1. After computing the $L_{2}$-sensitivity of $\Lambda_{j}$ for j=0,1, and 2, we can now compute the noise array $e_{j}∼N\left(0,\tau_{j}^{2}\right)$, where $\tau_{j}=\frac{\Delta_{j}}{\epsilon }\sqrt{2\log\frac{1.25}{\delta }}$, and then compute $\left\{{\hat{\Lambda }}_{j}\right\}$ following (Equation 5). Using these, we can compute the (ϵ,δ) differentially-private ${\hat{f}}_{D}\left(w\right)$ according to (Equation 6), and consequently, the minimizer ${\hat{w}}^{*}={arg min}_{w}{\hat{f}}_{D}\left(w\right)$. Again we note that the $L_{1}$-sensitivity of the coefficients in FM [7] is $\Delta^{fm}=\frac{1}{N}\left(\frac{D^{2}}{4}+3D\right)$ and the $L_{2}$-sensitivity of the coefficients in relaxed FM [10] is $\Delta^{rlx-fm}=\frac{1}{N}\sqrt{\frac{D^{2}}{16}+D}$ for logistic regression. As in the case of linear regression, both of these sensitivities are orders of magnitude larger than $\Delta_{j}$ that we achieved for j∈{1,2}, and for practical values of *D* and *N*. Since additive noise variances of our proposed Gaussian FM do not depend on the sample dimension *D*, we obtain ${\hat{f}}_{D}\left(w\right)$, the (ϵ,δ)-differentially private approximation to ${\tilde{f}}_{D}\left(w\right)$, with much less noise. As mentioned before, we validate our analysis empirically using synthetic and real datasets in Section 6.

### 4.3. Avoiding Unbounded Noisy Objective Functions

Our proposed Gaussian FM achieves (ϵ,δ)-DP by injecting noise drawn from a Gaussian distribution into the coefficients of the Stone–Weierstrass decomposition of the empirical average objective function. However, the injection of noise may render the objective function *unbounded*, which means there may not exist any optimal solution for the noisy objective function. As shown in Section 4.1 and Section 4.2, the Stone–Weierstrass decomposition would transform the objective functions of linear and logistic regression problems into quadratic polynomials in our Gaussian FM. Let ${\hat{f}}_{D}\left(w\right)=w^{⊤}Mw+\alpha^{⊤}w+\beta$ be the matrix representation of the quadratic polynomial, where M is a symmetric and positive semi-definite matrix, α is a *D*-dimensional vector and β is a scalar. After injection of noise, the noisy objective function becomes ${\hat{f}}_{D}\left(w\right)=w^{⊤}\hat{M}w+{\hat{\alpha }}^{⊤}w+\hat{\beta }$. In order to ensure that ${\hat{f}}_{D}\left(w\right)$ is bounded after introducing noise, it suffices to make sure $\hat{M}$ is also symmetric and positive semi-definite [42].

We follow the seminal work of Dwork et al. [43] in our implementation—the symmetry of $\hat{M}$ is ensured by constructing the noise matrix in such a way that noise is first drawn from the Gaussian distribution to form an upper triangular matrix, and the elements of the upper triangle part of the matrix (excluding the diagonal elements) are then copied to its lower triangle part. Adding the symmetric noise matrix to M results in a symmetric $\hat{M}$. However, ${\hat{f}}_{D}\left(w\right)$ may still be unbounded if $\hat{M}$ is not positive semi-definite. To resolve this, we perform eigen-decomposition of $\hat{M}$ to obtain the eigenvalues and corresponding eigenvectors. We then project the eigenvalues onto the non-negative orthant. Let $Q^{⊤}SQ$ be the eigen-decomposition of $\hat{M}$, where Q is a D×D matrix containing an eigenvector of $\hat{M}$ in each row, and S is a diagonal matrix where the *i*-th diagonal element is the eigenvalue of $\hat{M}$ corresponding to the eigenvector in the *i*-th row of Q. We can write
$${\hat{f}}_{D}\left(w\right)=w^{⊤}\left(Q^{⊤}SQ\right)w+{\hat{\alpha }}^{⊤}w+\hat{\beta }.$$
If the *i*-th diagonal element of S is negative, we turn that entry to zero. After this projection onto the non-negative orthant, let the resulting matrix be $\hat{S}$, where any *i*-th diagonal element is bigger than or equal to zero. The noisy objective function then becomes
$${\hat{f}}_{D}\left(w\right)=w^{⊤}\left(Q^{⊤}\hat{S}Q\right)w+{\hat{\alpha }}^{⊤}w+\hat{\beta },$$
where $\left(Q^{⊤}\hat{S}Q\right)$ is symmetric positive semi-definite. Thus, ${\hat{f}}_{D}\left(w\right)$ is bounded. Since all of these are performed after the differentially-private noise addition, we can invoke the post-processing invariability of differential privacy and guarantee that ${\hat{f}}_{D}\left(w\right)$ is (ϵ,δ)-differentially private. Consequently, the minimizer ${\hat{w}}^{*}$ also satisfies (ϵ,δ) differential privacy. Note that it is possible for all the eigenvalues of the differentially private estimate of the M matrix to be negative. We leave the solution to such cases for future work.

## 5. Extension of Gaussian FM to Decentralized-Data Setting: capeFM

In many signal processing and machine learning applications, the privacy-sensitive user data being collected/used are of decentralized nature. Training machine learning and neural-network-based models on such a huge amount of data is certainly lucrative from an algorithmic perspective, but privacy constraints often make it challenging to share such datasets with a central aggregator. However, training locally at one node/site is infeasible due to the number of samples in each node/site could be too small for meaningful model training. Decentralized DP can benefit such research work by allowing data owners to share information while maintaining local privacy. The conventional decentralized DP scheme, however, always results in a degradation in performance compared to that of the pooled-data scenario. In this section, we first describe the problem with conventional decentralized DP. Then we review the CAPE scheme [6] in brief, as we employ the CAPE scheme into our Gaussian FM to propose capeFM.

**The Decentralized-data Setting.** In line with our discussions in Section 2.2, let us consider a decentralized data setting with *S* sites and a central aggregator node. We assume an “honest but curious” threat model [6]: all parties follow the protocol honestly, but a subset are “curious” and can collude (maybe with an external adversary) to learn other sites’ data/function outputs. For simplicity, we consider the symmetric setting: each site s∈[S] holds a dataset $D_{s}$ of $N_{s}=\frac{N}{S}$ disjoint data samples $\left(x_{s,n},y_{s,n}\right)$, where the total number of samples across all sites is *N*, and $x_{s,n}\in R^{D}$. The cost incurred by the model parameters $w\in R^{D}$ due to one data sample is $f\left(x_{s,n};w\right):R^{D}\times R^{D}\mapsto R$. We need to minimize the average cost to find the optimal $w^{*}$. The empirical average cost for a particular w over all the samples is expressed as
$$\begin{array}{cc} f_{D}\left(w\right) & =\frac{1}{N}\sum_{s=1}^{S}\sum_{n=1}^{N_{s}}f\left(x_{s,n};w\right)=\frac{1}{S}\sum_{s=1}^{S}\frac{1}{N_{s}}\sum_{n=1}^{N_{s}}f\left(x_{s,n};w\right). \end{array}$$
According to (Equation 3), the above expression can be written as
$$\begin{array}{cc} f_{D}\left(w\right) & =\frac{1}{S}\sum_{s=1}^{S}\sum_{j=0}^{J}\langle \Lambda_{j}^{s},{\bar{\phi }}_{j}\rangle =\sum_{j=0}^{J}\langle \Lambda_{j},{\bar{\phi }}_{j}\rangle , \end{array}$$
where $\Lambda_{j}^{s}$ contains $\frac{1}{N_{s}}\sum_{n=1}^{N_{s}}\lambda_{\phi s,n}$ as its entries for all $\phi \left(w\right)\in \Phi_{j}$ at site *s*, $\Lambda_{j}=\frac{1}{S}\sum_{s=1}^{S}\Lambda_{j}^{s}$, and ${\bar{\phi }}_{j}$ is the array containing all $\phi \left(w\right)\in \Phi_{j}$ as its entries. Finally, we can compute the minimizer:$$\begin{array}{cc} w^{*} & =\underset{w}{arg min}f_{D}\left(w\right)=\underset{w}{arg min}\sum_{j=0}^{J}\langle \Lambda_{j},{\bar{\phi }}_{j}\rangle . \end{array}$$

### 5.1. Problems with Conventional Decentralized DP Computations

In this section, we discuss the problems with conventional decentralized DP schemes [6]. Consider estimating the mean $f\left(x\right)=\frac{1}{N}\sum_{n=1}^{N}x_{n}$ of *N* scalars $x={\left[x_{1},\ldots ,x_{N-1},x_{N}\right]}^{⊤}$, where each $x_{n}\in \left[0,1\right]$. The $L_{2}$-sensitivity of the function f(x) is $\frac{1}{N}$. Therefore, for computing the (ϵ,δ)-DP estimate of the average a=f(x), we can follow the Gaussian mechanism [4] to release ${\hat{a}}_{pool}=a+e_{pool}$, where $e_{pool}∼N\left(0,\tau_{pool}^{2}\right)$ and $\tau_{pool}=\frac{1}{N\epsilon }\sqrt{2\log\frac{1.25}{\delta }}$.

Suppose now that the *N* samples are equally distributed among *S* sites. An aggregator wishes to estimate and publish the mean of all the samples. For preserving privacy, the conventional DP approach is for each site *s* to release (or send to the aggregator node) an (ϵ,δ)-DP estimate of the function $a_{s}=f\left(x_{s}\right)$ as: ${\hat{a}}_{s}=f\left(x_{s}\right)+e_{s}$, where $e_{s}∼N\left(0,\tau_{s}^{2}\right)$ and $\tau_{s}=\frac{1}{N_{s}\epsilon }\sqrt{2\log\frac{1.25}{\delta }}=\frac{S}{N\epsilon }\sqrt{2\log\frac{1.25}{\delta }}$. The aggregator can then compute the (ϵ,δ)-DP approximate average as
$${\hat{a}}_{conv}=\frac{1}{S}\sum_{s=1}^{S}{\hat{a}}_{s}=\frac{1}{S}\sum_{s=1}^{S}a_{s}+\frac{1}{S}\sum_{s=1}^{S}e_{s}.$$
The variance of the estimator ${\hat{a}}_{conv}$ is $S\cdot \frac{\tau_{s}^{2}}{S^{2}}=\frac{\tau_{s}^{2}}{S}≜\tau_{conv}^{2}$. We observe the ratio
$$\frac{\tau_{pool}^{2}}{\tau_{conv}^{2}}=\frac{\tau_{s}^{2}/S^{2}}{\tau_{s}^{2}/S}=\frac{1}{S}.$$
That is, the decentralized DP averaging scheme will always result in a poorer performance than the pooled-data case. Imtiaz et al. [6] proposed the CAPE protocol that improves the performance of such systems by assuming the availability of some reasonable resources.

### 5.2. Correlation Assisted Private Estimation (CAPE)

**Trust/Collusion Model.** In order to incorporate the CAPE scheme to our proposed Gaussian FM in a decentralized data setting, we assume a similar trust model as in [6]. As mentioned before, we assume all of the *S* sites and the central aggregator node to be honest-but-curious. That is, the sites and central node can collude with an adversary to learn about the data or function output of some other site. We assume that up to $S_{C}=\left\lceil \frac{S}{3}\right\rceil -1$ sites, as well as the central node can collude with an adversary. In addition to having access to the outputs from each site and the aggregator, the adversary can know everything about the $S_{C}$ colluding sites, including their private data. Denoting the non-colluding sites with $S_{H}$, we have $S=S_{C}+S_{H}$.

**Correlated Noise and the CAPE Protocol.** Imtiaz et al. [6] proposed a novel framework that ensures (ϵ,δ)-DP guarantee of the output from each site, while achieving the same noise level of the pooled-data scenario in the final output from the aggregator. In the CAPE scheme, each site s∈[S] first generates two noise terms: $g_{s}∼N\left(0,\tau_{g}^{2}\right)$ locally, and $e_{s}∼N\left(0,\tau_{e}^{2}\right)$ jointly with all other sites such that $\sum_{s=1}^{S}e_{s}=0$. The correlated noise term $e_{s}$ is generated by employing the secure aggregation protocol (SecureAgg) by Bonawitz et al. [28], which utilizes Shamir’s *t*-out-of-*n* secret sharing [44] and is communication-efficient. The procedure is outlined in Algorithm 2.
**Algorithm 2** Generate Zero-Sum Noise**Require:** Local noise variances $\left\{\tau_{s}^{2}\right\}$; security parameter λ; threshold value *t*1:Each site generates ${\hat{e}}_{s}∼N\left(0,\tau_{s}^{2}\right)$2:Aggregator computes $\sum_{s=1}^{S}{\hat{e}}_{s}$ according to SecureAgg(λ,t) [28]3:Aggregator broadcasts $\sum_{s=1}^{S}{\hat{e}}_{s}$ to all sites s∈[S]4:Each site computes $e_{s}={\hat{e}}_{s}-\frac{1}{S}\sum_{s^{\prime }=1}^{S}{\hat{e}}_{s^{\prime }}$5:**return** 
$e_{s}$

Note that neither of the terms $e_{s}$ and $g_{s}$ has large enough variance to provide an acceptable (ϵ,δ)-DP guarantee. However, the variances of $e_{s}$ and $g_{s}$ are chosen in such a way that the noise $e_{s}+g_{s}$ is sufficient to ensure a stringent DP guarantee to $f(x_{s})$ at site *s*. We observe that the variance of $e_{s}$ is given by $\tau_{e}^{2}=\left(1-\frac{1}{S}\right)\tau_{s}^{2}$ and the variance of $g_{s}$ is set to $\tau_{g}^{2}=\frac{\tau_{s}^{2}}{S}$ [6]. Considering the decentralized mean computation problem of Section 5.1, under the CAPE scheme, each site sends ${\hat{a}}_{s}=f\left(x_{s}\right)+e_{s}+g_{s}$ to the aggregator. We can then compute the following at the aggregator
$$a_{cape}=\frac{1}{S}\sum_{s=1}^{S}{\hat{a}}_{s}=\frac{1}{S}\sum_{s=1}^{S}f\left(x_{s}\right)+\frac{1}{S}\sum_{s=1}^{S}g_{s},$$
where we used $\sum_{s=1}^{S}e_{s}=0$. The variance of the estimator $a_{cape}$ is $\tau_{cape}^{2}=S\cdot \frac{\tau_{g}^{2}}{S^{2}}=\tau_{pool}^{2}$, which is exactly the same as if all the data were present at the aggregator. This claim is formalized in Lemma 1 [6] in Section 2.1. That is, the CAPE protocol achieves the same noise variance as the pooled-data scenario in the symmetric decentralized-data setting.

### 5.3. Proposed Gaussian FM for Decentralized Data (capeFM)

For employing the CAPE scheme to extend our proposed Gaussian FM for decentralized-data setting, we need to generate the zero-sum noise. We can readily extend Algorithm 2 to generate array-valued zero-sum noise terms for each of the $\Lambda_{j}$ terms of the decomposition (Equation 3). That is, according to the CAPE scheme, the sites generate the noise $e_{j}^{s}$ using Algorithm 2, such that $\sum_{s=1}^{S}e_{j}^{s}=0$ holds for all j∈{0,…,J}. The sites also generate noise $g_{j}^{s}$ with entries i.i.d. $∼N(0,{\tau_{jg}^{s}}^{2})$. The sites then compute the perturbed coefficient arrays locally as ${\hat{\Lambda }}_{j}^{s}=\Lambda_{j}^{s}+e_{j}^{s}+g_{j}^{s}$ for all j∈{0,…,J} and send ${\hat{\Lambda }}_{j}^{s}$ to the central aggregator. Note that $e_{j}^{s}$ and $g_{j}^{s}$ are arrays of the same dimension as $\Lambda_{j}^{s}$. Now, the aggregator simply computes the average of each coefficient term for all j∈{0,…,J} as
$$\begin{array}{cc} {\hat{\Lambda }}_{j} & =\frac{1}{S}\sum_{s=1}^{S}{\hat{\Lambda }}_{j}^{s}=\frac{1}{S}\sum_{s=1}^{S}\Lambda_{j}^{s}+\frac{1}{S}\sum_{s=1}^{S}g_{j}^{s}, \end{array}$$
because $\sum_{s}e_{j}^{s}=0$. The aggregator then uses these $\left\{{\hat{\Lambda }}_{j}\right\}$ to compute ${\hat{f}}_{D}\left(w\right)=\sum_{j=0}^{J}\langle {\hat{\Lambda }}_{j},{\bar{\phi }}_{j}\rangle$ and release ${\hat{w}}^{*}={arg min}_{w}{\hat{f}}_{D}\left(w\right)$. The privacy of capeFM follows directly from Theorem 1 and Theorem 2. It follows from Lemma 1 [6] that in the symmetric setting (i.e., $N_{s}=\frac{N}{S}$ and $\tau_{j}^{s}=\tau_{j}$ for all sites s∈[S] and all j∈{0,1,…,J}), the noise variance achieved at the aggregator is the same as that of the pooled-data scenario. Additionally, the performance gain of capeFM over any conventional decentralized functional mechanism is given by Proposition 4. We refer to our proposed decentralized functional mechanism as capeFM, shown in Algorithm 3.
**Algorithm 3** Proposed Decentralized Gaussian FM (capeFM)**Require:** Data samples $\left(x_{s,n},y_{s,n}\right)$ for s∈[S]; cost function $f_{D}\left(w\right)$ as in (Equation 3); local noise variances $\left\{\tau_{j}^{2}\right\}$ for all j∈{0,…,J}  1:**for** 
0≤s≤S 
**do**  2:    **for** 0≤j≤J **do**  3:        Compute $\Lambda_{j}^{s}$ as shown in Section 4  4:        Generate $e_{j}^{s}$ according to Algorithm 2 (entrywise)  5:        Compute ${\tau_{jg}^{s}}^{2}=\frac{{\tau_{j}^{s}}^{2}}{S}$  6:        Generate $g_{j}^{s}$ with entries i.i.d. $∼N(0,{\tau_{jg}^{s}}^{2})$  7:        Compute ${\hat{\Lambda }}_{j}^{s}=\Lambda_{j}^{s}+e_{j}^{s}+g_{j}^{s}$  8:    **end for**  9:**end for**10:At the central aggregator, compute for all j∈{0,…,J}: ${\hat{\Lambda }}_{j}=\frac{1}{S}\sum_{s=1}^{S}{\hat{\Lambda }}_{j}^{s}$11:Compute ${\hat{f}}_{D}\left(w\right)=\sum_{j=0}^{J}\langle {\hat{\Lambda }}_{j},{\bar{\phi }}_{j}\rangle$12:**return** Perturbed objective function ${\hat{f}}_{D}\left(w\right)$

### 5.4. Computation and Communication Overhead of capeFM

We analyze the computation and communication costs associated with the proposed capeFM algorithm according to [6,28] for the decentralized linear regression and logistic regression problems. At each iteration round, we need to generate the zero-sum noise terms $e_{j}^{s}$, which entails $O(S+D^{2})$ communication complexity of the sites and $O(S^{2}+SD^{2})$ communication complexity of the aggregator [28]. Each site computes the noisy coefficient arrays $\Lambda_{j}^{s}$ and sends those to the aggregator, incurring an $O(D^{2})$ communication cost for the sites. Therefore, the total communication cost is $O(S+D^{2})$ for the sites and $O(S^{2}+SD^{2})$ for the aggregator node. On the other hand, the zero-sum noise generation entails $O(S^{2}+SD^{2})$ computation cost at the sites and $O(S^{2}D^{2})$ computation cost at the aggregator [28]. This is expected since the largest coefficient arrays we are computing/sending are D×D matrices in the decentralized setting. Note that we are not incorporating the computation cost of ${\hat{w}}^{*}={arg min}_{w}{\hat{f}}_{D}\left(w\right)$.

## 6. Experimental Results

In this section, we empirically compare the performance of our proposed Gaussian FM algorithm (**gauss-fm**) with those of some state-of-the-art differentially private linear and logistic regression algorithms, namely noisy gradient descent (**noisy-gd**) [12], objective perturbation (**obj-pert**) [8], original functional mechanism (**fm**) [7], and relaxed functional mechanism (**rlx-fm**) [10]. We also compare the performance of these algorithms with non-private linear and logistic regression (**non-priv**). As mentioned before, we compute the overall ϵ using RDP for the multi-round **noisy-gd** algorithm. Additionally, we show how our proposed decentralized functional mechanism (**cape-fm**) can improve a decentralized computation if the target function has sensitivity satisfying the conditions of Proposition 5 in Section 2.1. We show the variation in performance with privacy parameters and number of training samples. For the decentralized setting, we further show the empirical performance comparison by varying the number of sites.

**Performance Indices.** For the linear regression task, we use the mean squared error (MSE) as the performance index. Let the test dataset be $D_{\mathrm{test}}=\left\{\left(x_{n},y_{n}\right)\in X\times Y:n\in \left[N_{\mathrm{test}}\right]\right\}$. Then the MSE can be defined as: $\mathrm{MSE}=\frac{1}{N_{\mathrm{test}}}\sum_{n=1}^{N_{\mathrm{test}}}{\left({\hat{y}}_{n}-y_{n}\right)}^{2}$, where ${\hat{y}}_{n}$ is the prediction from the algorithm. For the classification task, we use accuracy as the performance index. The accuracy can be defined as: $\mathrm{Accuracy}=\frac{1}{N_{\mathrm{test}}}\sum_{n=1}^{N_{\mathrm{test}}}I\left(\mathrm{round}\left({\hat{y}}_{n}\right)=y_{n}\right)$, where I(·) is the indicator function, and ${\hat{y}}_{n}$ is the prediction from the algorithm. Note that, in addition to a small MSE or large accuracy, we want to attain a strict privacy guarantee, i.e., small overall (ϵ,δ) values. Recall from Section 3 that the overall ϵ for multi-shot algorithms is a function of the number of iterations, the target δ, the additive noise variance $\tau^{2}$ and the $L_{2}$ sensitivity Δ. To demonstrate the overall ϵ guarantee for a fixed target δ, we plotted the overall ϵ (with dotted red lines on the right *y*-axis) along with MSE/accuracy (with solid blue lines on the left *y*-axis) as a means for visualizing how the privacy–utility trade-off varies with different parameters. For a given privacy budget (or performance requirement), the user can use the overall ϵ plot on the right *y*-axis, shown with dotted lines, (or MSE/accuracy plot on the left *y*-axis, shown with solid lines) to find the required noise standard deviation τ on the *x*-axis and, thereby, find the corresponding performance (or overall ϵ). We compute the overall ϵ for the **noisy-gd** algorithm using the RDP technique shown in Section 3.

### 6.1. Linear Regression

For the linear regression problem, we perform experiments on three real datasets (and a synthetic dataset, as shown in Appendix B). The *pharmacogenetic* dataset was collected by the *International Warfarin Pharmacogenetics Consortium* (**IWPC**) [23] for the purpose of estimating personalized warfarin dose based on clinical and genotype information of a patient. The data used for this study have ambient dimension D=9, and features are collected from N=5052 patients. Out of the wide variety of numerical modeling methods used in [23], linear regression provided the most accurate dose estimates. Fredrikson et al. [20] later implemented an attack model assuming an adversary who employed an inference algorithm to discover the genotype of a target individual, and showed that an existing functional mechanism (**fm**) failed to provide a meaningful privacy guarantee to prevent such attacks. We perform privacy-preserving linear regression on the **IWPC** dataset (Figure 1a–c) to show the effectiveness of our proposed **gauss-fm** over **fm**, **rlx-fm**, and other existing approaches. Additionally, we use the *Communities and Crime* dataset (**crime**) [45], which has a larger dimensionality D=101 (Figure 1d–f), and the *Buzz in Social Media* dataset (**twitter**) [46] with D=77 and a large sample size N=10,000 (Figure 1g–i). We refer the reader to [47] for a detailed description of these real datasets. For all the experiments, we pre-process the data so that the samples satisfy the assumptions $∥x_{n}{∥}_{2}\leq 1$ and $y_{n}\in \left[-1,1\right]$∀n∈[N]. We divide each dataset into train and test partitions with a ratio of 90:10. We show the average performance over 10 independent runs.

**Performance Comparison with Varying τ.** We first investigate the variation of MSE with the DP additive noise standard deviation τ. We plot MSE against τ in Figure 1a,d,g. Recall from Definition 3 that, in the Gaussian mechanism, the noise is drawn from a Gaussian distribution with standard deviation $\tau =\frac{\Delta }{\epsilon }\sqrt{2\log\frac{1.25}{\delta }}$. We keep δ fixed at $10^{-5}$. Note that one can vary ϵ to vary τ. Since noise standard deviation is inversely proportional to ϵ, increasing ϵ means decreasing τ, i.e., smaller noise variance. We observe from the plots that smaller τ leads to smaller MSE for all DP algorithms, indicating better utility at the expense of higher privacy loss. It is evident from these MSE vs. τ plots that our proposed method **gauss-fm** has much smaller MSE compared to all the other methods for the same τ values for all datasets. The **obj-pert** and **fm** algorithms offer pure DP by trading off utility, whereas **gauss-fm** and **rlx-fm** algorithms offer approximate DP. Although **rlx-fm** improves upon **fm**, the excess noise due to linear dependence on data dimension *D* leads to higher MSE than **gauss-fm**. Our proposed **gauss-fm** outperforms all of these methods by reducing the additive noise with the novel sensitivity analysis as shown in Section 4. We recall that the overall privacy loss for **noisy-gd** is calculated using the RDP approach, since noise is injected into the gradients in every iteration during optimization, with target $\delta =10^{-5}$. On the other hand, **gauss-fm**, **rlx-fm**, and **fm** add noise to the polynomial coefficients of the cost function $f_{D}\left(w\right)$ before optimization, and **obj-pert** injects noise into the regularized cost function [8]. We plot the total privacy loss for all of the algorithms against τ. We observe from the *y*-axis on the right that the total privacy loss of the multi-round **noisy-gd** is considerably higher than the single-shot algorithms.

**Performance Comparison with Varying $N_{train}$.** Next, we investigate the variation of MSE with the number of training samples $N_{train}$. For this task, we shuffle and divide the total number of samples *N* into smaller partitions and perform the same pre-processing steps, while keeping the test partition untouched. We kept the values of the privacy parameters fixed: ϵ=0.5 and $\delta =10^{-5}$. We plot MSE against $N_{train}$ in Figure 1b,e,h. We observe that performance generally improves with the increase in $N_{train}$, which indicates that it is easier to ensure the same level of privacy when the training dataset cardinality is higher. We also observe from the MSE vs. $N_{train}$ plots that our proposed method **gauss-fm** offers MSE very close to that of **non-priv** even for moderate sample sizes, outperforming **fm**, **rlx-fm**, **noisy-gd**, and **obj-pert**. Again, we compute the overall ϵ spent using RDP for **noisy-gd**, and show that the multi-round algorithm suffers from larger privacy loss. Recall from (Equation 7) in Section 3 that the overall ϵ depends on sensitivity Δ, and the number of iterations *T*. In the computation of $\frac{\tau^{2}}{\Delta^{2}}$, the number of training samples $N_{train}$ is cancelled out. Thus, the overall ϵ depends only on *T* for **noisy-gd**. We keep *T* fixed at 1000 iterations for **noisy-gd** and observe that the overall privacy risk exceeds 20. Note that we set the value of the target $\delta_{r}$ in (Equation 7) to be equal to δ in our computations.

**Performance Comparison with Varying δ.** Recall that we can interpret the privacy parameter δ as the probability that an algorithm fails to provide privacy risk ϵ. The **obj-pert** and **fm** algorithms offer pure ϵ-DP, where the additional privacy parameter δ is zero. Hence, we compare our proposed **gauss-fm** method with the **rlx-fm** and **noisy-gd** methods, which also guarantee (ϵ,δ)-DP. In the Gaussian mechanism, δ is in the denominator of the logarithmic term within the square root in the expression of τ. Therefore, the noise variance $\tau^{2}$ is not significantly changed by varying δ. We keep privacy parameter ϵ fixed at 0.5 and observe from the MSE vs. δ plots in Figure 1c,f,i show that the performance of our algorithm does not degrade much for smaller δ. For the **IWPC** dataset in Figure 1c, for a value of δ as small as $10^{-2}$ (indicating 1% probability of the algorithm failing to provide ϵ-differential privacy), the MSE of **gauss-fm** is almost the same as that of the **non-priv** case. For the other datasets, our proposed method also gives better performance and overall ϵ, and thus a better privacy–utility trade-off than **rlx-fm** and **noisy-gd**.

### 6.2. Logistic Regression

For the logistic regression problem, we again perform experiments on three real datasets (and a synthetic dataset, as shown in Appendix B): the *Phishing Websites* dataset (**phishing**) [47] with dimensionality D=30 (Figure 2a–c), the *Census Income* dataset (**adult**) [47] with D=13 (Figure 2d–f), and the *KDD Cup ’99* dataset (**kdd**) [47] with D=36 (Figure 2g–i). As before, we pre-process the data so that the feature vectors satisfy $∥x_{n}{∥}_{2}\leq 1$, and $y_{n}\in \left\{0,1\right\}$∀n∈[N]. Note for **obj-pert** that the cost function is regularized and the labels are assumed to be $\left\{-1,1\right\}$ in [8]. We divide each dataset into train and test partitions with a ratio of 90:10. We use percent accuracy on the test dataset as the performance index for logistic regression, and show the average performance over 10 independent runs.

**Performance Comparison with Varying τ.** We plot accuracy against the DP additive noise standard deviation τ in Figure 2a,d,g. We observe that accuracy degrades when the additive DP noise standard deviation τ increases, indicating a greater privacy guarantee at the cost of performance. When noise is too high, privacy-preserving logistic regression may not learn a meaningful w at all, and provide random results. Depending on the class distribution, this may not be obvious and the accuracy score may be misleading. We observe this for the **kdd** dataset in Figure 2g, where the classes are highly imbalanced, with ∼80% positive labels. Although the existing **fm** performs poorly on this dataset, our proposed **gauss-fm** provides significantly higher accuracy for all datasets, outperforming **fm**, as well as **rlx-fm**, **obj-pert**, and **noisy-gd**. As before, we observe the total privacy loss, i.e., overall ϵ spent, from the *y*-axis on the right.

**Performance Comparison with Varying $N_{train}$.** We perform the same steps described in Section 6.1 and observe the variation in performance with the number of training samples, $N_{train}$ while keeping the privacy parameters fixed in Figure 2b,e,h. Accuracy generally improves with increasing $N_{train}$. We observe that the same DP algorithm does not perform equally well for different datasets. For example, **obj-pert** performs better than **noisy-gd** on the **adult** dataset (Figure 2e), whereas **noisy-gd** performs better than **obj-pert** on the **phishing** dataset (Figure 2b). In general, **fm** and **rlx-fm** suffer from too much noise due to the quadratic and linear dependence on *D* of their sensitivities, respectively. However, our proposed **gauss-fm** overcomes this issue and consistently achieves accuracy close to the **non-priv** case even for moderate sample sizes. We also show the overall privacy guarantee, as before.

**Performance Comparison with Varying δ.** Similar to the linear regression experiments shown in Section 6.1, we keep ϵ and $N_{train}$ fixed for this task and vary the other privacy parameter δ. Figure 2c,f,i show that percent accuracy improves with increased δ. For sufficiently large δ (indicating 1–5% probability of the algorithm failing to provide ϵ privacy risk), **gauss-fm** accuracy can reach that of the **non-priv** algorithm in some datasets (e.g., Figure 2i). Although the accuracy of **noisy-gd** also improves, it comes at the cost of additional privacy risk, as shown in the overall ϵ vs. δ plots along the *y*-axes on the right. Due to the higher noise variance, **rlx-fm** achieves much inferior accuracy compared to both **gauss-fm** and **noisy-gd**.

### 6.3. Decentralized Functional Mechanism (capeFM)

In this section, we empirically show the effectiveness of capeFM, our proposed decentralized Gaussian FM which utilizes the CAPE [6] protocol. We implement differentially private linear and logistic regression for the decentralized-data setting using the same datasets described in Section 6.1 and Section 6.2, respectively. Note that the IWPC [23] data were collected from 21 sites across 9 countries. After obtaining informed consent to use de-identified data from patients prior to the study, the Pharmacogenetics Knowledge Base has since made the dataset publicly available for research purpose. As mentioned before, the type of data contained in the IWPC dataset is similar to many other medical datasets containing private information [20].

We implement our proposed **cape-fm** according to Algorithm 3, along with **fm**, **rlx-fm**, **obj-pert**, and **noisy-gd** according to the conventional decentralized DP approach. We compare the performance of these methods in Figure 3 and Figure 4. Similar to the pooled-data scenario, we also compare performance of these algorithms with non-private linear and logistic regression (**non-priv**). For these experiments, we assume $N_{s}=\frac{N}{S}$ and $\tau_{s}=\tau$. Recall that the CAPE scheme achieves the same noise variance as the pooled-data scenario in the symmetric setting (see Lemma 1 [6] in Section 2.1). As our proposed capeFM algorithm follows the CAPE scheme, we attain the same advantages. When varying privacy parameters and $N_{train}$, we keep the number of sites *S* fixed. Additionally, we show the variation in performance due to change in the number of sites in Figure 5. We pre-process each dataset as before, and use MSE and percent accuracy on test dataset as performance indices of the decentralized linear and logistic regression problems, respectively.

**Performance Comparison by Varying τ.** For this experiment, we keep the total number of samples *N*, privacy parameter δ, and the number of sites *S* fixed. We observe from the plots (a), (d), and (g) in both Figure 3 and Figure 4 that as τ increases, the performance degrades. The proposed **cape-fm** outperforms conventional decentralized **noisy-gd**, **obj-pert**, **fm**, and **rlx-fm** by a larger margin than the pooled-data case. The reason for this is that we can achieve a much smaller noise variance at the aggregator due to the correlated noise scheme detailed in Section 5.3. The utility of **cape-fm** thus stays the same as the centralized case in the decentralized-data setting, whereas the conventional scheme’s utility always degrades by a factor of *S* (see Section 5.1). The overall ϵ usage vs. τ plots on the right y-axes for each site show that **noisy-gd** suffers from much higher privacy loss.

**Performance Comparison by Varying $N_{train}$.** We keep ϵ, δ, and *S* fixed while investigating variation in performance with respect to $N_{train}$. As the sensitivities we computed in Section 4.1 and Section 4.2 are inversely proportional to the sample size, it is straightforward to infer that guaranteeing smaller privacy risk and higher utility is much easier when the sample size is large. Similar to the pooled-data cases in Section 6.1 and Section 6.2, we again observe from the plots (b), (e), and (h) in both Figure 3 and Figure 4 that, for sufficiently large $N_{train}=SN_{s,train}$, utility of **cape-fm** can reach that of the **non-priv** case. Note that the **non-priv** algorithms are the same as the pooled-data scenario, because if privacy is not a concern, all sites can send the data to aggregator for learning.

**Performance Comparison by Varying δ.** For this task, we keep ϵ, $N_{train}$, and *S* fixed. Note according to the CAPE scheme that the proposed **cape-fm** algorithm guarantees (ϵ,δ)-DP where (ϵ,δ) satisfy the relation $\delta =2\frac{\sigma_{z}}{\epsilon -\mu_{z}}\phi \left(\frac{\epsilon -\mu_{z}}{\sigma_{z}}\right)$. Recall that δ is the probability that the algorithm fails to provide privacy risk ϵ, and that we assumed a fixed number of colluding sites $S_{C}=\left\lceil \frac{S}{3}\right\rceil -1$. From the plots (c), (f), and (i) in both Figure 3 and Figure 4, we observe that even for moderate values of δ, **cape-fm** easily outperforms **rlx-fm** and **noisy-gd**. Moreover, as seen from the overall ϵ plots, **noisy-gd** provides a much weaker privacy guarantee. Thus, our proposed **cape-fm** algorithm offers superior performance and privacy–utility trade-off in the decentralized setting.

**Performance Comparison by Varying *S*.** Finally, we investigate performance variation with the number of sites *S*, keeping the privacy and dataset parameters fixed. This automatically varies the number of samples $N_{s}$ at each site s∈[S], as we consider the symmetric setting. Figure 5a–c shows the results for decentralized linear regression, and Figure 5d–f shows the results for decentralized logistic regression. We observe that the variation in *S* does not affect the utility of **cape-fm**, as long as the number of colluding sites meets the condition $S_{C}\leq \left\lceil \frac{S}{3}\right\rceil -1$. However, increasing *S* leads to significant degradation in performance for conventional decentralized DP mechanisms, since the additive noise variance increases as $N_{s}$ decreases. We show additional experimental results on synthetic datasets in Appendix B.

## 7. Conclusions and Future Work

In this paper, we proposed Gaussian FM that offers a significant improvement over the existing FM to compute functions that are commonly used in signal processing and machine learning applications, satisfying differential privacy. Our improvement stems from a novel sensitivity analysis that resulted in an orders-of-magnitude reduction in the amount of noise added to the coefficients of the Stone–Weierstrass decomposition of the functions. We showed two common regression problems—linear and logistic regression—as examples to demonstrate our analyses. Additionally, we experimentally showed the superior privacy guarantee and utility of our proposed method over existing methods by varying privacy parameters and relevant dataset parameters for both synthetic and real datasets. We extended our Gaussian FM algorithm to decentralized data settings by taking advantage of a correlated noise protocol, CAPE, and proposed capeFM, which ensures the same utility as the pooled-data scenario in certain regimes. We empirically compared the performance of the proposed capeFM with that of existing and conventional algorithms for decentralized linear and logistic regression problems. In addition to varying privacy and dataset parameters, we showed performance comparison by varying the number of sites, which further proves the superior privacy guarantee and improved utility of our proposed method. For future work, we plan to extend our research to more complex algorithms and neural networks to ensure differential privacy on other challenging signal processing and machine learning problems.

## Figures and Tables

**Figure 1 entropy-25-00825-f001:**
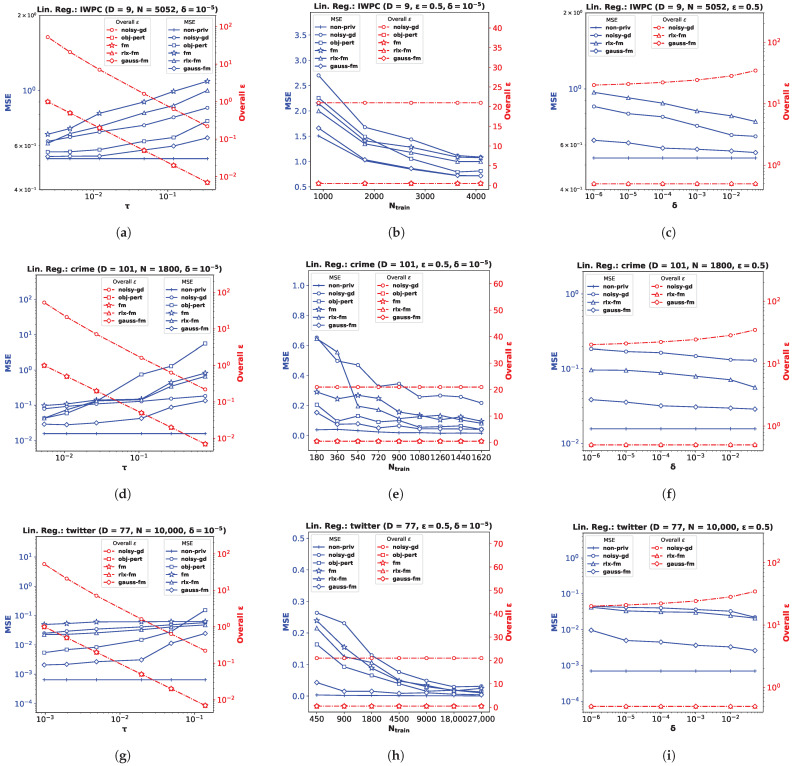
Linear regression performance comparison in terms of MSE and overall ϵ for *IWPC* (D=9), *crime* (D=101), and *twitter* (D=77) datasets with varying noise standard deviation τ in (**a**,**d**,**g**) the number of training samples $N_{train}$ in (**b**,**e**,**h**), and privacy parameter δ in (**c**,**f**,**i**).

**Figure 2 entropy-25-00825-f002:**
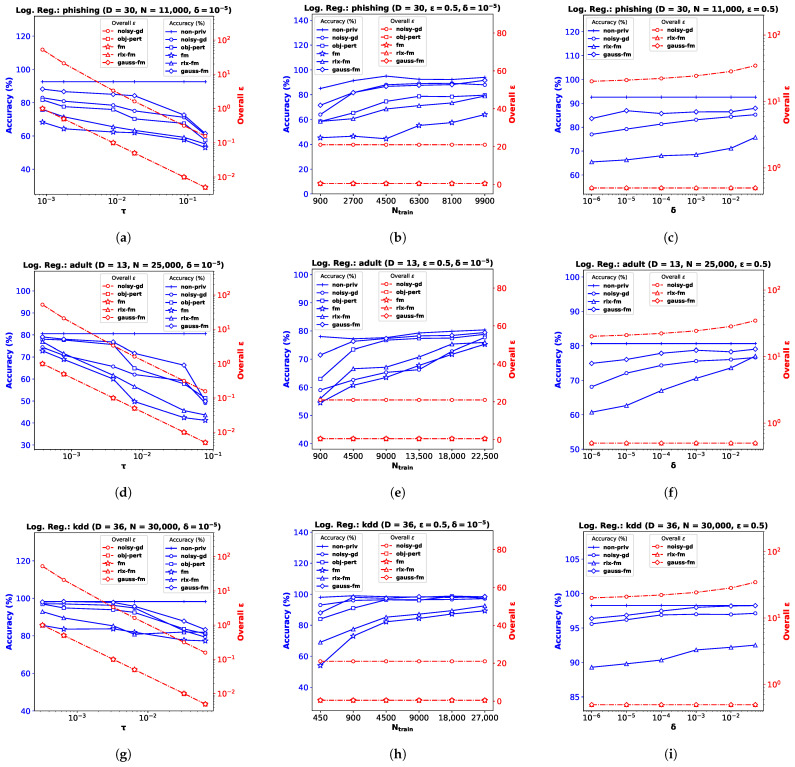
Logistic regression performance comparison in terms of accuracy and overall ϵ for *phishing* (D=30), *adult* (D=13), and *kdd* (D=36) datasets with varying noise standard deviation τ in (**a**,**d**,**g**), the number of training samples $N_{train}$ in (**b**,**e**,**h**), and privacy parameter δ in (**c**,**f**,**i**).

**Figure 3 entropy-25-00825-f003:**
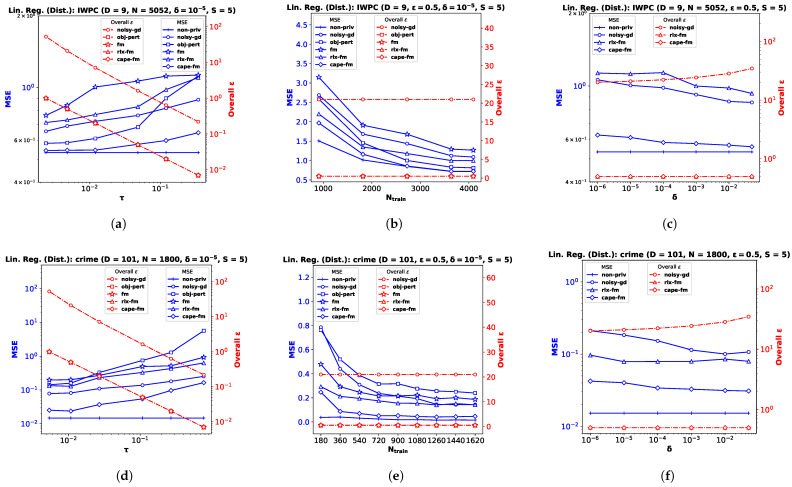
Decentralized linear regression performance comparison in terms of MSE and overall ϵ for *IWPC* (D=9), *crime* (D=101), and *twitter* (D=77) datasets with varying noise standard deviation τ in (**a**,**d**,**g**), the number of training samples $N_{train}$ in (**b**,**e**,**h**), and privacy parameter δ in (**c**,**f**,**i**).

**Figure 4 entropy-25-00825-f004:**
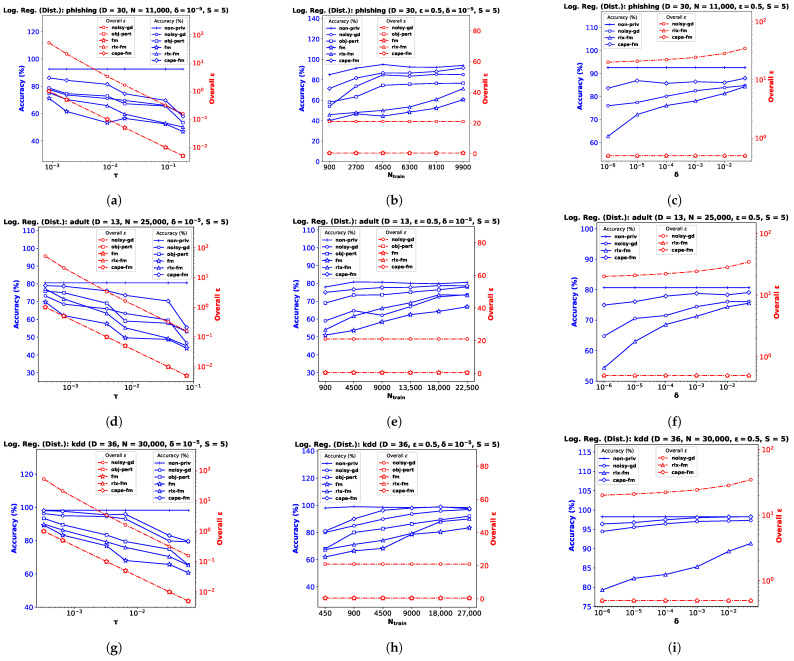
Decentralized logistic regression performance comparison in terms of accuracy and overall ϵ for *phishing* (D=30), *adult* (D=13), and *kdd* (D=36) datasets with varying noise standard deviation τ in (**a**,**d**,**g**), the number of training samples $N_{train}$ in (**b**,**e**,**h**), and privacy parameter δ in (**c**,**f**,**i**).

**Figure 5 entropy-25-00825-f005:**
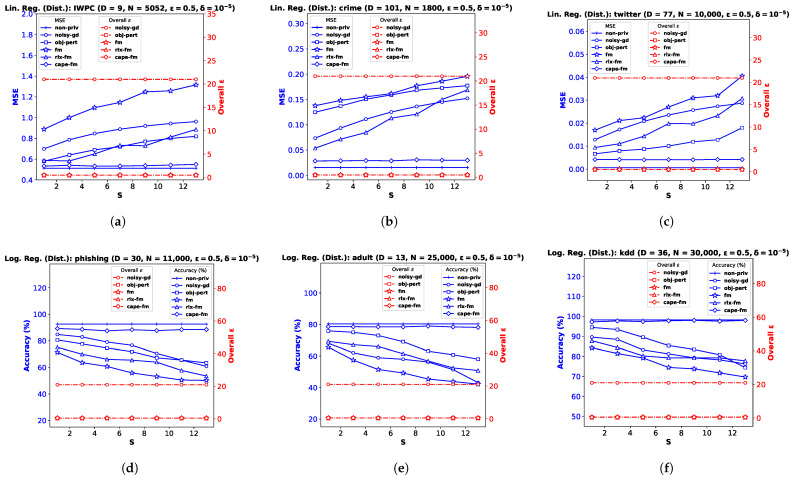
Decentralized linear and logistic regression performance comparison and overall ϵ with varying number of sites *S* for the datasets (**a**) *IWPC* (D=9), (**b**) *crime* (D=101), (**c**) *twitter* (D=77), (**d**) *phishing* (D=30), (**e**) *adult* (D=13), and (**f**) *kdd* (D=36).

## Data Availability

The experimental data used to evaluate the performance of the algorithms proposed in this paper are available from the corresponding author upon request.

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
