# Peer review of "Approximating Functions with Approximate Privacy for Applications in Signal Estimation and Learning"

_entropy, 2023, doi:10.3390/e25050825_

Round 1

Reviewer 1 Report

The paper studies an interesting problem and the results found would be of interest to some audience.

My main issue is the novelty and contribution of this work. The main improvement in this paper is using the Gaussian mechanism instead of the Laplace mechanism, thus changing the pure DP into approximate DP. It makes sense that having approximate DP instead of pure DP would give better results. Therefore, it does not seem like a significant improvement in terms of technique.

The paper would also require a light proofreading for the typos, as some sentences would be better modified to make them more understandable, as well as some minor typos.

Moreover, it would be a good idea to have more explanation about CAPE in the bulk of the paper rather than appendix, if possible. 

One more thing about the results is that the figures are not color-blind friendly, and maybe that is one thing to consider for the revision.

Reviewer 2 Report

This paper proposed a Gaussian mechanism based functional mechanism. Similar idea was attempted in Ding, et al, but not cited.

The difference in this paper and Ding, et al are (1) different noise sigma. Ding follows the extended Gaussian mechanism which is improved up on the original Gaussian mechanism by Dwork. Simply comparing Algorithm 1 in this paper and Algorithm 2 in Ding's work, Ding's proposal add less noise for given epsilon, delta. The author should compare with Ding's mechanism theoretically and empirically. (2) The author also used RDP to further tighter bound the epsilon. This is a novel improvement over Ding's work. 

Ding, J., Zhang, X., Li, X., Wang, J., Yu, R., & Pan, M. (2020). Differentially Private and Fair Classification via Calibrated Functional Mechanism. Proceedings of the AAAI Conference on Artificial Intelligence, 34(01), 622-629. https://doi.org/10.1609/aaai.v34i01.5402

In the experiment, it is not a fair comparison between original FM and Gaussian FM, since Gaussian FM is achieving relaxed DP. Ding's FM should be directly compared against the proposed Gaussian FM here.

The author also proposed to use CAPE for Gaussian FM in the decentralized setting. The method is technically sound. A missing related work is using FM for vertically partitioned setting. 

D. Xu, S. Yuan and X. Wu, "Achieving Differential Privacy in Vertically Partitioned Multiparty Learning," 2021 IEEE International Conference on Big Data (Big Data), Orlando, FL, USA, 2021, pp. 5474-5483, doi: 10.1109/BigData52589.2021.9671502.

In the decentralized experiment, the computational overhead is not evaluated.

The author need to address the above issues to be accepted.

Round 2

Reviewer 1 Report

I thank the authors for their feedback and updates. I think the paper is okay now for acceptance.

Reviewer 2 Report

The author addressed the comments in latest version. I recommend to accept this work.